# Airborne Extractive Electrospray Mass Spectrometry Measurements of the Chemical Composition of Organic Aerosol

Demetrios Pagonis[1,2], Pedro Campuzano-Jost[1,2], Hongyu Guo[1,2], Douglas A. Day[1,2], Melinda K. Schueneman[1,2], Wyatt L. Brown[1,2], Benjamin A. Nault[1,2,†], Harald Stark[1,2,3], Kyla Siemens[4], Alex Laskin[4], Felix Piel[5,6], Laura Tomsche[7,8], Armin Wisthaler[6,9], Matthew M. Coggon[2,10], Georgios I. Gkatzelis[2,10,§], Hannah S. Halliday[8,‡], Jordan E. Krechmer[3], Richard H. Moore[8], David S. Thomson[11], Carsten Warneke[2,10], Elizabeth B. Wiggins[8], and Jose L. Jimenez[1,2]

[1]Department of Chemistry, University of Colorado, Boulder, CO, USA
[2]Cooperative Institute for Research in Environmental Sciences (CIRES), University of Colorado, Boulder, CO, USA
[3]Aerodyne Research Inc., Billerica, MA, USA
[4]Department of Chemistry, Department of Earth, Atmospheric and Planetary Sciences, Purdue University, West Lafayette, Indiana, USA
[5]IONICON Analytik GmbH, Innsbruck, Austria
[6]Institut für Ionenphysik und Angewandte Physik, Universität Innsbruck, Innsbruck, Austria
[7]Universities Space Research Association, Columbia, MD, USA
[8]NASA Langley Research Center, Hampton, VA, USA
[9]Department of Chemistry, University of Oslo, Oslo, Norway
[10]National Oceanic and Atmospheric Administration Chemical Sciences Laboratory, Boulder, CO, USA
[11]Original Code Consulting, Boulder, CO, USA
[†]Now at: Aerodyne Research, Inc., Billerica, MA, USA
[‡]Now at: Office of Research and Development, US EPA, Research Triangle Park, NC
[§]Now at: Institute of Energy and Climate Research, IEK-8: Troposphere, Forschungszentrum Jülich GmbH, Jülich, Germany.

*Correspondence to*: Jose L. Jimenez (jose.jimenez@colorado.edu)

**Abstract.** We deployed an extractive electrospray ionization time-of-flight mass spectrometer (EESI-MS) for airborne measurements of biomass burning aerosol during the Fire Influence on Regional to Global Environments and Air Quality (FIREX-AQ) study onboard the NASA DC-8 research aircraft. Through optimization of the electrospray working solution, active control of the electrospray region pressure, and precise control of electrospray capillary position, we achieved 1 Hz quantitative measurements of aerosol nitrocatechol and levoglucosan concentrations up to pressure altitudes of 7 km. EESI-MS response to levoglucosan and nitrocatechol was calibrated for each flight, with flight-to-flight calibration variability of 60% (1σ). Laboratory measurements showed no aerosol size dependence in EESI-MS sensitivity below particle geometric diameters of 400 nm, covering 82% of accumulation mode aerosol mass during FIREX-AQ. We also present a first in-field intercomparison of EESI-MS with a chemical analysis of aerosol online proton-transfer-reaction mass spectrometer (CHARON PTR-MS) and a high-resolution Aerodyne aerosol mass spectrometer (AMS). EESI-MS and CHARON PTR-MS levoglucosan concentrations were well correlated, with a regression slope of 0.94, $R^2$ = 0.77. AMS levoglucosan-equivalent concentrations and EESI-MS levoglucosan showed greater difference, with a regression slope of 1.36, $R^2$ = 0.96, likely indicating the contribution of other compounds to the AMS levoglucosan-equivalent measurement. Total EESI-MS signal

showed correlation ($R^2 = 0.9$) with total organic aerosol measured by AMS, and the EESI-MS bulk organic aerosol sensitivity was 60% of the sensitivity to levoglucosan standards.

## 1 Introduction

Extractive electrospray ionization time-of-flight mass spectrometry (EESI-TOF-MS, hereafter EESI-MS) allows for rapid measurements of the chemical composition of organic aerosol (OA) (Lopez-Hilfiker et al. 2019; Chen et al. 2006; Doezema et al. 2012). EESI-MS has been used to characterize sources of primary and secondary OA in cities (Stefenelli et al. 2019; Qi et al. 2019; Brown et al. 2020), track OA chemistry in laboratory studies (Doezema et al. 2012; Gallimore and Kalberer 2013; Gallimore et al. 2017; Liu et al. 2019a; Liu et al. 2019b), and proof-of-concept has been demonstrated for airborne applications (Lopez-Hilfiker et al. 2019).

During EESI-MS measurements, aerosol inlet flow is intercepted by an electrospray, where collisions of the aerosol particles with electrospray droplets lead to dissolution of particulate matter in the charged droplet, followed by droplet evaporation, ionization of the dissolved components (Kumbhani et al. 2018; Law et al. 2010), and detection by a high-resolution time-of-flight mass spectrometer (Junninen et al. 2010). The advantage of EESI-MS is the lack of sample preparation – analytes are not collected onto a vaporizing element or filter, allowing many compounds to be sensitively detected without thermal decomposition (Lopez-Hilfiker et al. 2019; Stark et al. 2017). Droplets are transferred into a vacuum through a steel capillary (residence time = 1.8 ms) that is heated to 250 °C, which facilitates droplet evaporation. The soft ionization of EESI-MS allows for quantitative measurements of individual compounds, providing insight into the chemical pathways involved in the formation and evolution of ambient OA that is more detailed and source-specific than what can be achieved with harsher ionization techniques (Qi et al. 2019; Stefenelli et al. 2019; Tong et al. 2020).

Two key parameters that determine the range of compounds detectable with EESI-MS are the composition of the electrospray solution and the ion polarity. EESI-MS sensitivity has been shown to vary by orders of magnitude based on the solubility of analytes in the electrospray solution (Law et al. 2010). Previous EESI-MS measurements of ambient OA have utilized positive mode (EESI(+)) with sodium iodide electrospray dopant facilitating detection of many compounds as sodium adducts $[M+Na]^+$ (Lopez-Hilfiker et al. 2019; Stefenelli et al. 2019; Qi et al. 2019; Brown et al. 2020), while negative ion polarity (EESI(−)) has been employed in several laboratory studies of OA composition, using acetic acid or formic acid as electrospray dopants to detect deprotonated analytes $[M-H]^-$ (Chen et al. 2006; Gallimore and Kalberer 2013). EESI(−) has also been used in ambient measurements of metals in aerosol using ethylenediamine tetraacetic acid (EDTA) to detect chelated metals $[EDTA+X]^-$ (Giannoukos et al. 2019).

Airborne measurements of OA concentration and composition have been carried out by: filters for offline analysis (Maria et al. 2002; Huebert et al. 2004; Heald et al. 2005; Forrister et al. 2015); particle-into-liquid sampler (PILS) coupled to a total organic carbon analyzer (Sullivan et al. 2006; Duong et al. 2011); Aerodyne aerosol mass spectrometer (AMS) (DeCarlo et al. 2008); particle analysis by laser mass spectrometry (PALMS) (Froyd et al. 2019; Murphy et al. 1998); and chemical analysis

of aerosol online with proton-transfer-reaction mass spectrometry (CHARON PTR-MS) (Piel et al. 2019). PILS coupled to

offline ion chromatography (PILS-IC) (Sullivan et al. 2014, 2019) and CHARON PTR-MS (Piel et al. 2019) have both

quantified levoglucosan in biomass burning OA from airborne platforms, with PILS-IC demonstrating a detection limit of 0.1

ng m$^{-3}$ at a 2-minute sampling frequency, and CHARON PTR-MS demonstrating 4 ng m$^{-3}$ detection limits at 1 Hz sampling

and 0.5 ng m$^{-3}$ with 2-minute averaging. To our knowledge, no airborne measurements of nitrocatechol, another major

component of biomass burning OA (Iinuma et al. 2010; Finewax et al. 2018), have been reported.

We deployed EESI-MS in a configuration that allowed for quantitative detection of components of biomass burning OA at

pressure altitudes up to 7 km during the Fire Influence on Regional to Global Environments and Air Quality (FIREX-AQ)

study onboard the National Aeronautics and Space Administration (NASA) DC-8 aircraft. This was achieved by optimizing

the electrospray solution for performance at pressures suitable for airborne sampling, development of an automated

electrospray capillary stage, and extensive flight-day and in-flight calibrations with a colocated AMS. Here we describe the

instrument adaptations and its performance as deployed during FIREX-AQ and present comparisons to AMS and CHARON

PTR-MS measurements during that campaign.

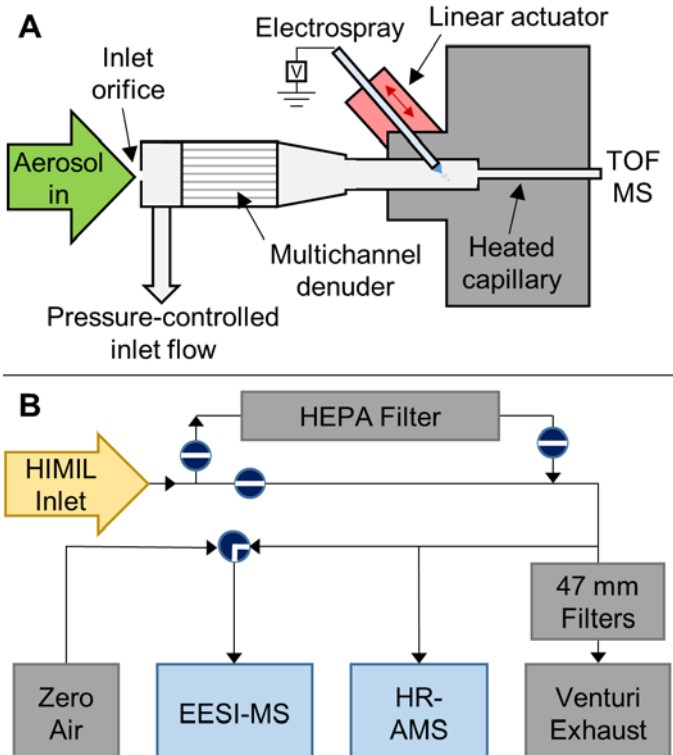

**Figure 1.** (A) Schematic of the EESI-MS source, pressure-controlled inlet, and automated capillary stage; and (B) EESI-MS and HR-AMS
sampling configuration flown during FIREX-AQ. The linear actuator controlling capillary position is opposed by a spring (not drawn) to
allow for bidirectional control of capillary position, shown by red arrows. The valves in (B) are drawn in the positions used for ambient
sampling.

## 2 Experimental Section

### 2.1 Instrument description

The sample flow path of the EESI-MS deployed for this study (Aerodyne Research, Inc. Billerica, MA, USA) is shown in Fig. 1. The National Center for Atmospheric Research (NCAR) High-Performance Instrumented Airborne Platform for Environmental Research Modular Inlet (HIMIL) (NCAR EOL 2019; Stith et al. 2009) is shared with the University of Colorado high-resolution AMS (DeCarlo et al. 2006; Canagaratna et al. 2007; Nault et al. 2018; Guo et al. 2020). The AMS and EESI-MS shared several inlet components: a high-efficiency particulate air (HEPA) filter (Pall Corp., Port Washington, NY, USA)

for removal of ambient aerosol when measuring instrument backgrounds and quantifying detection limits; a calibration system for monodisperse aerosol consisting of an atomizer (TSI 3076, Shoreview, MN, USA), differential mobility analyzer (TSI 3081), and condensation particle counter (CPC; TSI 3010); and a polydisperse aerosol generation system consisting of a medical nebulizer (deVilbiss, Somerset, PA, USA) operated with ultra-high purity zero air (Praxair, Danbury, CT, USA) at 1.4 bar.

The EESI-MS pressure-controlled inlet (PCI) contains the multichannel activated carbon denuder and the electrospray capillary (Fig. 1). Air enters the PCI through a 350 μm flat-plate platinum orifice (Ladd Research, Williston, VT, USA), and exits the PCI through both the mass spectrometer and a pump (KNF Neuberger, Inc., Trenton, NJ, USA), with the flow rate of air through the pump modulated by a pressure controller (Alicat Scientific, Tucson, AZ, USA). When the DC-8 reached the operational ceiling of the EESI-MS PCI during high-altitude transits, the inlet of the PCI was automatically switched from

ambient air to either UHP zero air or filtered air from the aircraft cabin. This provided a source of air at sufficiently high pressure to ensure that the PCI pressure never dropped below the set point, a necessary condition for maintaining stable electrospray. Establishing and calibrating a new electrospray while airborne takes time, thereby reducing data coverage, and so loss of spray was avoided whenever possible. All automated valves, pressure controllers, and data logging for instrument flows, pressures, and temperatures were controlled using the MICAS-X software (Original Code Consulting, Boulder, CO,

USA) in a LabVIEW environment (NI, Austin, TX, USA).

    The instrument background — signal attributable to the electrospray itself or to contaminants in the ionization chamber — was measured for 15 seconds every 3 minutes by switching the PCI inlet from ambient air to UHP zero air. Time response of EESI-MS to these background measurements was about 5 s, as shown in Fig. S1. Background signals were linearly interpolated between measurements. The instrument detection limits were then determined against this background by sampling ambient

air through the main inlet HEPA filter, which was done for 15 seconds every 18 minutes. Detection limits were calculated for each filter period and interpolated across ambient sampling.

    Organic gases in the atmosphere are detectable by secondary electrospray ionization (SESI) (Zhao et al. 2017), and so must be removed from the sample flow when measuring organic aerosol by EESI-MS. The denuder used to strip away organic gases from the sample air in this study is an extruded activated carbon cylinder 3.2 cm long and 1.6 cm in diameter, with

approximately 300 square channels. The denuder was regenerated by baking at 90 °C in a flow of dry zero air for 8 h after

each flight. The denuder efficiency in removing gas-phase compounds is demonstrated in Fig. 2 by the comparison of biomass burning plumes sampled using EESI(−) with and without the denuder present. When the denuder is present (Fig. 2A, Fig. S2A), EESI-MS acetate signal ($C_2H_3O_2^-$) increases by less than 100 counts $s^{-1}$ during the intercept of a plume with approx. 60 ppb acetic acid. With the denuder removed (Fig. 2B), an EESI-MS acetate signal exceeding 104 counts $s^{-1}$ is observed during an intercept of a similarly concentrated plume, indicating a denuder efficiency of over 99% for acetic acid. The significant tailing in the EESI-MS acetate signal is consistent with partitioning delays expected for small organic molecules in a metal inlet (Liu et al. 2019c; Deming et al. 2019).

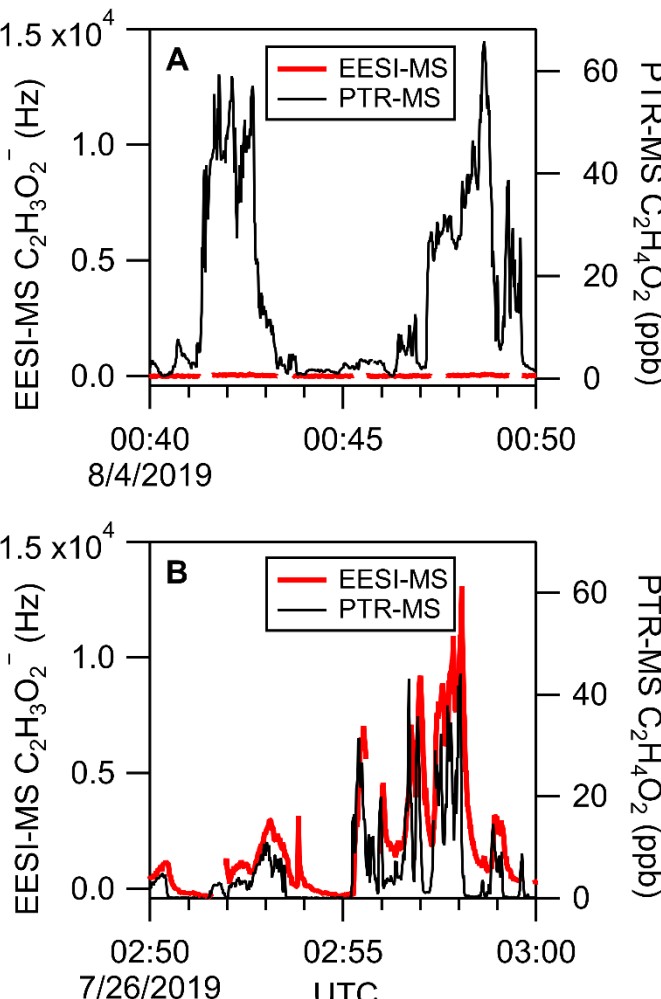

**Figure 2.** Demonstration of EESI-MS denuder efficiency for removing gas-phase VOCs. (A) EESI(−) acetate signal during wildfire smoke sampling with the carbon denuder in the inlet and (B) with no denuder in place. Comparisons to PTR-MS measurements of $C_2H_4O_2$ (predominantly acetic acid) is included to show that similar concentrations of gas-phase acetic acid were sampled in both flight segments. The same figure in panel (A) with a different Y-scaling that shows all the detail of EESI(−) acetate signal is shown in Fig. S2.

Inlet residence times and transmission efficiency were calculated across DC-8 sampling altitudes using the geometry of the inlet tubing and the flow rates used. Calculation of transmission efficiency accounts for particle losses from gravitational settling, impaction, diffusion, and aspiration. Total EESI-MS inlet residence times range from 1.4–1.6 s, and are shown as a function of sampling altitude, PCI pressure, and inlet subassembly in Fig. S3. Over half of the residence time is due to the volume of the PCI, which was designed to ensure laminar flow at the entrance and exit of the denuder. The calculated transmission efficiency of the inlet is shown as a function of sampling altitude in Fig. S4 and is separated by loss process in Fig. S5. The efficiency is calculated to be above 90% for particle geometric diameters between 50 and 350 nm, with 50% transmission at roughly 15 nm and 1 μm, depending on flight altitude. Particle volume distributions were measured by a laser aerosol spectrometer (LAS; Model 3340A, TSI, St. Paul, MN, USA) operated by the NASA Langley Aerosol Research Group, and the campaign-average particle volume distribution showed that 95% of aerosol volume was in particles with optical diameters between 100 and 460 nm. The LAS optical size range was calibrated using electrical mobility-classified, dry ammonium sulfate aerosols (refractive index of 1.52-0$i$). EESI-MS inlet transmission is calculated to be constant within 5% in that size range, as shown in Fig. S6.

Organic gases are removed by the denuder during sampling to prevent gas-phase ionization by SESI. The removal of semivolatile gases disturbs gas-particle equilibrium, potentially leading to aerosol evaporation inside the inlet. Heating of ambient air as it flows through inlet tubing also drives aerosol evaporation. We calculate upper limits for the extent of evaporative losses as a function of saturation vapor concentration at 298 K ($C*_{298}$) using a volatility basis set for biomass burning OA (May et al. 2013), ideal gas-particle partitioning (Pankow 1994; Donahue et al. 2006), and the kinetic evaporation model of Cappa (2010). We assume that particle evaporation is irreversible with no recondensation once aerosol enters the denuder, and with no kinetic limitations due to the aerosol phase state, to make the calculated evaporative losses an upper limit. The residence time from the entrance of the denuder is 0.65 s, and we calculate that levoglucosan and nitrocatechol ($C*_{298}$ = 13 μg m$^{-3}$ for both; May et al. 2012; Finewax et al. 2018) undergo losses of under 2% inside the inlet for all plume conditions sampled. Evaporation is calculated to be greater for higher-volatility compounds. During FIREX-AQ the total OA evaporation while sampling smoke is estimated as 0-28%, with evaporation being greatest when both OA concentrations and the temperature difference between the DC-8 cabin and ambient air were high (we note that OA evaporation is significantly lower in the AMS inlet, as the inlet residence time is a factor of three shorter than that of the EESI-MS). The plume transect estimated to undergo 28% OA evaporation had an OA concentration of 1760 μg sm$^{-3}$ and a $\Delta T$ of 34 K. The evaporative loss is estimated to be almost entirely due to compounds with $C*_{298}$ > 10$^4$ μg m$^{-3}$, three orders of magnitude more volatile than levoglucosan and nitrocatechol, which are estimated to undergo evaporation of under 2% in these conditions.

The electrospray capillary position was controlled using a linear stepper motor (Thorlabs ZFS13, Newton, NJ, USA) opposed by a 9 N spring. A photograph of the custom stage is included in the Supplement (Fig. S7). The stage gives the operator sub-mm precision in capillary position, allowing optimization of the electrospray even in turbulent flight conditions. We find that the electrospray capillary position where the primary ESI signal is greatest is also the position where EESI-MS signal is greatest (Fig. S8), allowing the user to optimize the electrospray capillary position without use of online aerosol standards. We interpret

this as an indication that the volume of the aerosol flow is larger than that of the electrospray, and that adjustments in electrospray capillary position are optimizing the extent to which the electrospray (and thereby the extracted and ionized aerosol components) is sampled by the aspiration of the mass spectrometer, rather than lost to ionization chamber walls. This suggests that improvements in EESI-MS sensitivity may be possible by narrowing the diameter of the electrospray region or focusing the aerosol upstream of the electrospray. The heated capillary is $4 \pm 0.5$ mm from the electrospray tip, depending on the optimized electrospray position. It is possible that future work optimizing this distance (and thereby time for droplet evaporation) may assist in achieving stable electrospray at lower pressures than those used in this study.

Ions produced by EESI are detected using an atmospheric pressure interface time-of-flight mass spectrometer (Junninen et al. 2010). Aerosol components that were not ionized by EESI are not focused by the ion optics of the TOF-MS and are pumped away or deposited on an internal surface of the ionization volume or mass spectrometer. The TOF-MS was operated at an extraction frequency of 21 kHz, recording up to $m/z = 700$. Any ions above this $m/z$ are recorded by the detector as part of a subsequent mass spectrum, in an effect known as "TOF wraparound" (Brown et al. 2020). Spectra were recorded and analyzed at 1 Hz throughout FIREX-AQ. During EESI(+) measurements resolving power ($m/\Delta m$) at $m/z$ 185 (levoglucosan) was 3,900. During EESI($-$) measurements resolving power at $m/z$ 154 (nitrocatechol) was 3,800. High-resolution mass spectrometric analysis was carried out in Tofware (Tofwerk AG, Thun, Switzerland and Aerodyne Research, Billerica, MA, USA), using purpose-built instrument diagnostic and analysis routines. These routines were automated for in-flight viewing of high-resolution time series.

## 2.2 Electrospray working solutions

The EESI(+) working solution used in this study was 3:1 methanol:water doped with 100 ppm NaI, leading to analyte detection as sodium ion adducts [M+Na]$^+$. The EESI($-$) working solution used was 3:1 methanol:water doped with 0.1% ($v/v$) formic acid, leading to analyte detection as deprotonated anions [M-H]$^-$. Chemical purities and suppliers are listed in the SI. Electrospray capillaries and the high-voltage electrode were cleaned with methanol prior to entering the working solutions to avoid any contamination, which would have increased instrument background over the course of the campaign. Working solutions were kept sealed to prevent evaporation from affecting the solvent:solute ratio.

The previous study that demonstrated EESI(+) was suitable for airborne applications utilized a 1:1 methanol:water working solution doped with 100 ppm NaI, and reported data up to a pressure altitude of 3 km (Lopez-Hilfiker et al. 2019). Increasing the methanol fraction of the working solution allows for more stable electrospray at decreasing electrospray region pressure, and this study's EESI-MS operated successfully at a pressure altitude of 7 km. At low pressures, heat transfer to an evaporating droplet is slower than at ambient pressure, and evaporative cooling of the electrospray droplets slows down their evaporation and can lead to droplets freezing (Marginean et al. 2009). Our 3:1 methanol:water working solution allowed for stable electrospray at pressures as low as 360 mbar, while a 1:1 methanol:water working solution was unstable below 700 mbar. We interpret this result as an indication that electrospray droplets from the 1:1 methanol:water solution were not evaporating fast

enough to produce ions upstream of the ion optics. Instead, Coulomb explosion of these droplets likely happened at some downstream location where resulting ions could not be efficiently focused and detected by the mass spectrometer.

The non-linear effect of decreasing electrospray region pressure on the efficiency of EESI is shown in Fig. 3A, where the EESI-MS sensitivity is reduced 83% when PCI pressure is reduced 30%, from 667 mbar to 467 mbar. An additional 23% reduction in pressure to 360 mbar results in 30% reduction in sensitivity. There are at least two separate processes contributing to the decrease in sensitivity: lower PCI pressure reducing the flow rate (and therefore mass flux) of aerosol into the mass spectrometer (given that the volumetric flow rate is constant), and the reduction of ESI ionization efficiency at low pressures discussed above. We include the contribution of the reduced flow rate in Fig. 3A, showing that it is the reduction in ionization efficiency that drives the non-linear relationship between electrospray region pressure and EESI-MS sensitivity.

These data indicate that small deviations in the electrospray region pressure can have substantial impacts on EESI-MS sensitivity. From the relationship shown in Fig. 3A, we calculate that a 25 mbar reduction in electrospray region pressure (e.g. 667 mbar to 642 mbar) can cause a 10% reduction in EESI-MS sensitivity. Pressure fluctuations of that magnitude are not unique to aircraft sampling: common sources of inlet pressure variability, such as pressure drops from sampling through particle filters or switching a valve, can approach 25 mbar. These fluctuations must be avoided during all EESI-MS measurements in order to avoid measurement bias from the pressure dependence of EESI-MS sensitivity. The electrospray region pressure during filter blanks and zero air backgrounds during FIREX-AQ was kept constant by the pressure controller. Pressure transients caused by valve switching were small (<20 mbar) and were stabilized within 2 s. Data acquired during these pressure transients were excluded from analysis.

The relationship between PCI pressure and EESI-MS sensitivity presented here is for a 3:1 methanol:water working solution, and similar reductions in sensitivity at lower pressures were also observed for acetonitrile:water working solutions. Measurement of the pressure dependence of EESI-MS sensitivity using a 1:1 methanol:water working solution was not achievable, as this solution did not give sufficiently stable spray at reduced pressure to allow for reliable calibration. A higher methanol fraction in the working solution could give better performance at low pressures than the 3:1 methanol:water solution used here, but as the 3:1 working solution showed suitable performance at the pressures relevant to FIREX-AQ this was not explored as part of this study. Changes to working solution composition can also have significant impacts on the extraction and ionization efficiency of particular components, and the linearity of EESI-MS response (Lopez-Hilfiker et al. 2019). It is therefore necessary to do extensive characterization of each new working solution tested. For example, a 3:1 acetonitrile:water working solution was tested and found to give stable electrospray at 467 mbar and linear response to varying analyte concentration. However, ionization of levoglucosan was found to be very inefficient in this solution and so it was deemed not suitable for use during FIREX-AQ, and was not characterized further.

There is significant potential for further investigation and optimization of electrospray dopants for EESI-MS. While use of NaI as an EESI(+) dopant provided sufficiently stable electrospray for 8 h research flights, a more volatile salt such as ammonium iodide may result in less salt deposition on the electrospray capillary and in the electrospray region. This could lead to more stable EESI(+) operation in situations where days of continuous electrospray are needed, such as ambient sampling at a ground

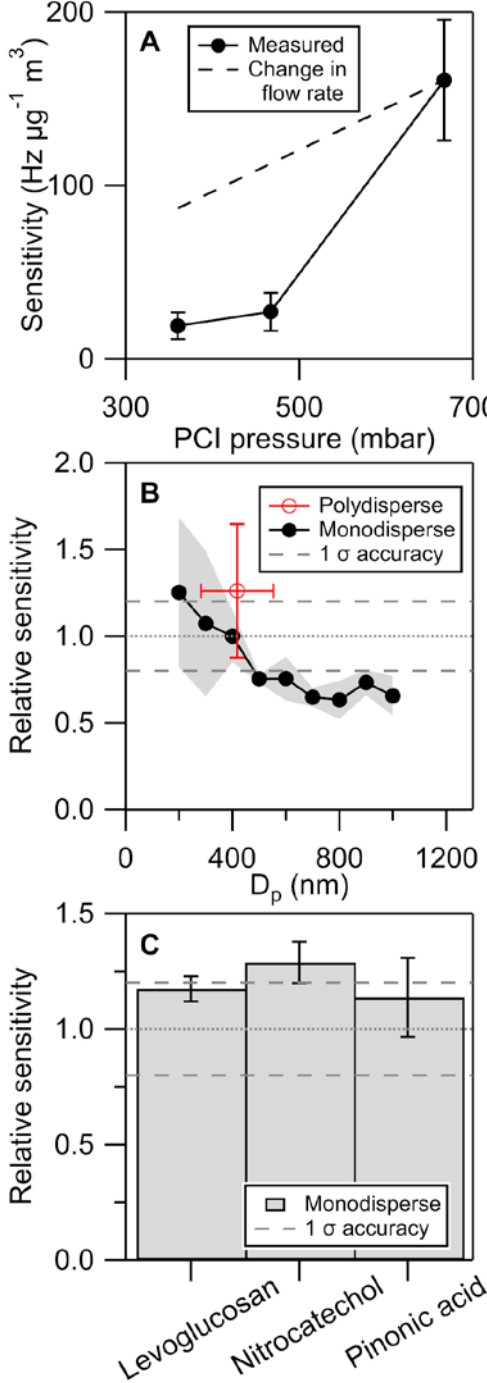

**Figure 3.** (A) Pressure dependence of EESI(−) nitrocatechol sensitivity , (B) particle diameter dependence of EESI(+) and EESI(−) sensitivity for all calibrants run during FIREX-AQ, scaled to sensitivity at 400 nm , and (C) EESI-MS sensitivities of pure compounds relative to sensitivities in a 50% mol/mol mixture. Levoglucosan was mixed with ammonium sulfate and analyzed using EESI(+). Nitrocatechol and pinonic acid were mixed with each other and analyzed using EESI(−).

site. The use of an acid dopant in negative-polarity electrospray has the potential to suppress the ionization of compounds less acidic than the dopant. As part of this study both formic acid and acetic acid were tested as dopants for EESI(−) and were
found to give similar sensitivity for nitrocatechol, despite a difference in acidity between the two dopants. It is possible that higher sensitivity to weakly acidic compounds could be achieved with a more weakly acidic dopant, or no dopant at all.

## 2.3 Calibrations

EESI-MS was calibrated against the AMS before and after each flight during FIREX-AQ using levoglucosan (EESI(+) calibrations) or 4-nitrocatechol (EESI(−) calibrations) standards aerosolized using a medical nebulizer. During maintenance
245  days EESI-MS was calibrated against the AMS and a CPC using monodisperse aerosol size-selected by a DMA. Prior studies have investigated the size-dependence of EESI-MS sensitivity using polydisperse aerosol, where the mode diameter of the size distribution was varied from 60–230 nm (Lopez-Hilfiker et al. 2019). EESI-MS calibrations using monodisperse aerosol have not been published to our knowledge; this is at least partially due to the change in inlet pressure (and hence sensitivity, cf. Section 2.2) imposed by most monodisperse particle generation systems and the lack of inlet pressure control in previous
studies. The size-dependence of the EESI-MS sensitivity to monodisperse aerosol is presented in Fig. 3B, averaging EESI(+) calibrations of levoglucosan, 4-nitrocatechol, ammonium nitrate, pinonic acid, and a 1:1 mixture of levoglucosan and ammonium sulfate. Sensitivities are normalized to 400 nm to allow inclusion of multiple calibrants with different sensitivities and to correct for day-to-day variability in EESI-MS sensitivity. The mean and standard deviation of flight-day polydisperse calibrations are also shown in Fig. 3B. The mode geometric diameter of the volume distribution during these calibrations
averaged 390 nm, measured by the AMS efficient particle time-of-flight (ePToF) mode. The decrease in EESI-MS sensitivity at particle diameters larger than 400 nm may be due to the particles becoming comparable to or larger than the droplets produced by the electrospray (Kumbhani et al. 2018). A similar mechanism may be responsible for the increase in EESI-MS sensitivity observed for 200 nm diameter particles. LAS measurements showed that for the average in-smoke FIREX-AQ particle volume size distribution the mode diameter was 300 nm, and 82% of the particle volume was in particles with diameters
below 400 nm (95% below 500 nm). AMS ePToF aerosol volume distribution measurements also showed an average mode geometric diameter of 300 nm, with 72% of particle volume at geometric diameters below 400 nm. Because the FIREX-AQ size distributions were mostly in the range where EESI-MS sensitivity shows minimal size dependence, we do not apply any particle size corrections to ambient EESI-MS data.

We estimate the uncertainty in the EESI-MS polydisperse calibration (2σ) to be 47%. This includes the variability between
replicate calibrations using the same electrospray hours apart (σ = 20%), the uncertainty in the AMS quantification (σ = 10%), and uncertainty in the EESI-MS transmission efficiency relative to the AMS (σ ≈ 10%). The day-to-day variability of the EESI-MS calibration factors (σ = 60%) is greater than the variability of calibrations done on a single continuously-operating electrospray (σ = 20%), showing the importance of calibrating EESI-MS after each new electrospray is established. Recalibration is necessary even if all conditions seem unchanged, as the same primary ESI ion signal can arise from
electrosprays with different properties. EESI-MS was completely powered off and left under vacuum at the end of each day,

necessitating the establishment of a new electrospray for every FIREX-AQ flight. The uncertainty in the AMS quantification of an aerosol standard is lower than the uncertainty reported for ambient OA ($\sigma$ = 19% for FIREX-AQ), since the product of the collection efficiency and relative ionization efficiency can be determined with high accuracy for aerosol standards (<10%) using both mass and single-particle calibrations (Xu et al. 2018; Hodshire et al. 2019).

The effect of the aerosol matrix on EESI-MS sensitivity was tested by nebulizing binary mixtures of analytes, size-selecting 300 nm particles with a DMA, and calibrating the EESI-MS against particle mass calculated from CPC counts, particle diameter, and the densities and mass fractions of the pure calibrants. EESI(+) matrix effects were investigated with a 1:1 mixture of levoglucosan and ammonium sulfate, and EESI(−) matrix effects were investigated with a binary mixture of 4-nitrocatechol and pinonic acid. Results of these investigations are shown in Fig. 3C, and show a potential 14–28% impact of

particle matrix on EESI-MS sensitivity, which is within the variability observed for replicate calibrations with a single electrospray ($\sigma$ = 20%). Here we assumed ideal mixing between the two components of each mixture, and the slight bias observed might be due to non-ideal mixing increasing the density of mixed particles.  Here we only tested mixtures that could be generated from a single nebulized aqueous solution, but previous studies have examined the effect of coatings on EESI-MS sensitivity and reported differing results (Lopez-Hilfiker et al. 2019; Kumbhani et al. 2018). It is discussed in Kumbhani et al.

(2018) that large particle size (up to 600 nm) may be a key factor in the incomplete solvation of multiphase aerosol particles, which would be consistent with the suppression of EESI-MS sensitivity observed in this study for particles with diameters larger than 400 nm. Additional studies are needed to separate the contributions of particle diameter and particle phase separation to EESI solvation efficiency. The instrument intercomparisons during measurement of wildfire smoke aerosol presented below provide evidence that EESI-MS sensitivity calculated from one-component and two-component calibrant

mixtures can be applied to more complex matrices, and that there were no phase state limitations on EESI-MS quantification of BBOA during FIREX-AQ.

EESI-MS detection limits during FIREX-AQ were calculated from periodic measurements of ambient air that had all aerosol removed by a HEPA filter. At a PCI pressure of 667 mbar, average EESI(+) levoglucosan and EESI(−) nitrocatechol detection limits (1 Hz, 3$\sigma$) were 695 and 18 ng sm$^{-3}$. At a PCI pressure of 467 mbar, average levoglucosan and nitrocatechol 1 Hz

detection limits were 770 and   50 ng sm$^{-3}$. The substantially higher levoglucosan detection limit is the result of greater instrument background, with a median background signal equivalent to 2.1 μg sm$^{-3}$ of aerosol levoglucosan, a factor of 1,000 greater than the median nitrocatechol EESI(−) background-equivalent concentration of 2.5 ng sm$^{-3}$. The background levoglucosan signal is resolved from neighboring peaks, as shown in Fig. S9. The detection limits varied with the sampling history of the instrument, with higher detection limits observed following sustained sampling of biomass burning OA,

persisting for hours (Fig. S10). Histograms of the detection limits obtained at each PCI pressure are presented in Fig. S11. The previously reported EESI(+) detection limit for levoglucosan is 10.5 ng sm$^{-3}$ for 30 s of averaging (scaled from 9.1 ng m$^{-3}$ at Zurich pressure and 295 K) (Stefenelli et al. 2019; Lopez-Hilfiker et al. 2019). If one assumes that the detection limit scales according to counting statistics, this corresponds to a 1-s detection limit of 58 ng sm$^{-3}$. Our levoglucosan detection limit at 667 mbar is roughly a factor of 12 higher, partly due to the change in working solution composition, difference in aspiration flow

rate caused by difference in sampling pressure (960 mbar vs 667 mbar, a factor of 1.4), and with a major contribution due to the reduction in sensitivity with operating pressure (Fig. 3). The levoglucosan detection limits achieved here using EESI-MS are also higher than that reported by Sullivan et al. (2014) using a PILS-IC with a 2-minute sampling time (0.1 ng m$^{-3}$), demonstrating the tradeoff between highly time-resolved measurements and more specific chromatographic measurements.

## 3 Results and Discussion

**3.1 Measurement of biomass burning organic aerosol**

Airborne EESI-MS measurements of biomass burning organic aerosol (BBOA) were carried out onboard the NASA DC-8 aircraft from 22 July – 3 September 2019 as part of the FIREX-AQ study (campaign map shown in Fig. S12). Flights based out of Boise, Idaho typically sampled wildland fire BBOA above mountainous terrain, and the EESI-MS was operated at a PCI pressure of 467 mbar for most of these flights. Flights based out of Salina, Kansas primarily sampled BBOA from small

agricultural fires at lower altitudes, and so EESI-MS was operated at a PCI pressure of 667 mbar for these flights. We consistently switched ion polarities throughout the study, totalling 17 EESI(+) flights and 10 EESI(−) flights, including test and transit flights. Electrospray polarity was only changed between flights. During three research flights (July 25, 29, and 30) the EESI-MS was flown without a denuder (due to denuder damage and delay in obtaining a replacement), and so we do not report any data from those flights other than what is shown in Fig. 2. EESI-MS was flown with a denuder for all other research

flights.

EESI-MS data at FIREX-AQ covers 414 out of 538 plume transects (77%). Of those transects with no EESI-MS data, the majority (76 out of 124) are from the three research flights where EESI-MS was flown without a denuder. Excluding those flights, EESI-MS data covers 90% of plume transects. Four percent of FIREX-AQ plume transects occurred above the operational ceiling of the EESI-MS.

Raw and background-corrected EESI(+) and EESI(−) mass spectra of BBOA sampled during FIREX-AQ are presented in Fig. 4. Spectra were acquired up to $m/z$ 700, but binned spectral analysis (Zhang et al. 2019) showed no correlation with CO above $m/z$ 400, and so spectra are only shown to that point. The majority of the raw signal arises directly from the electrospray solution itself, as opposed to extractive electrospray ionization of aerosol analytes, as shown in Fig. 4. We categorize fitted high-resolution time-of-flight peaks as aerosol if the average 1 Hz signal-to-noise ratio is above 0.5 (Brown et al. 2020). When

sampling typical plume concentrations (OA ≈ 50 μg sm-3), aerosol accounts for 8% of total fitted EESI(+) signal, and 9% of total fitted EESI(−) signal, though much of the background is resolvable by the TOF-MS. Signal-to-background ratios calculated from the spectra in Fig. 4A and C are shown in Fig. S13. These high backgrounds make frequent measurement of EESI-MS background signals a necessity, in order to keep minor changes in the background from overwhelming the background-subtracted aerosol signal. Measurement of the Allan variance of key EESI(+) and EESI(−) peaks while flying

(Fig. S14) showed that the electrospray background evolves rapidly enough in flight that averaging longer than ~20 seconds does not improve signal-to-noise.

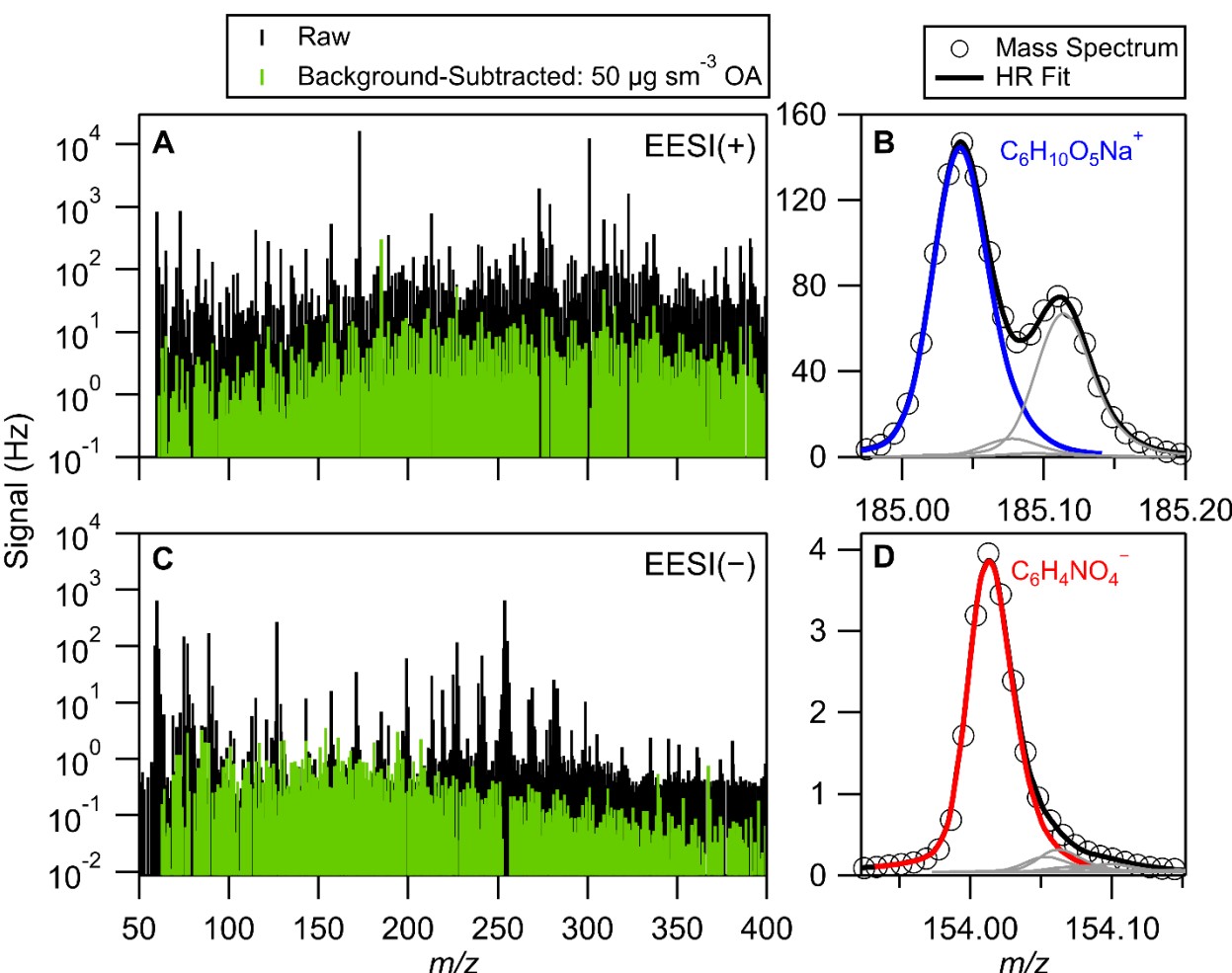

**Figure 4.** Raw and background-subtracted (A) EESI(+)and (C) EESI(−)spectra while sampling 50 µg sm$^{-3}$ of wildland fire smoke aerosol, and high-resolution mass spectra and peak fits of ions attributed to (B) levoglucosan and (D) nitrocatechol. The peaks shown in (B) and (D) are from the raw spectra in panels (A) and (C).

EESI(+) signal for the ion $C_6H_{10}O_5Na^+$ is attributed to anhydrohexoses and is referred to as "levoglucosan" here. Chromatographic studies have shown that levoglucosan comprises approximately 75% of anhydrosugars in biomass burning aerosol, with mannosan and galactosan (stereoisomers of levoglucosan) comprising the remainder (Sullivan et al. 2014). EESI(−) signal for the ion $C_6H_4NO_4^-$ is attributed to nitrocatechol, which is a major oxidation product of catechol (Finewax et al. 2018) — a primary emission from biomass burning (Koss et al. 2018). EESI-MS peak fitting for $C_6H_{10}O_5Na^+$ and $C_6H_4NO_4^-$ is shown in Fig. 4. To support these assignments, we collected aerosol onto 47 mm Teflon filters (Omnipore, Millipore Sigma, Burlington, MA, USA) during the study and analyzed filter extracts by HPLC-ESI-HRMS (Lin et al. 2018). The chromatogram of $C_6H_4NO_4^-$ consistently showed a single peak matching the retention time of a 4-nitrocatechol standard (Fig. S15), and the accurate measured mass confirmed the elemental assignment of the peak within 2 ppm mass accuracy at a resolving power

*m/Δm* of 100,000. It is possible that 3-nitrocatechol co-elutes with 4-nitrocatechol in the HPLC analysis, but since it has been
shown that 3-nitrocatechol yields from catechol oxidation are very low, we expect that 4-nitrocatechol is the dominant isomer
present in biomass burning OA (Finewax et al. 2018). Positive-ion HPLC-ESI-HRMS analysis also showed $C_6H_{10}O_5Na^+$ as a
single peak at the same retention time as a levoglucosan standard (Fig. S16).

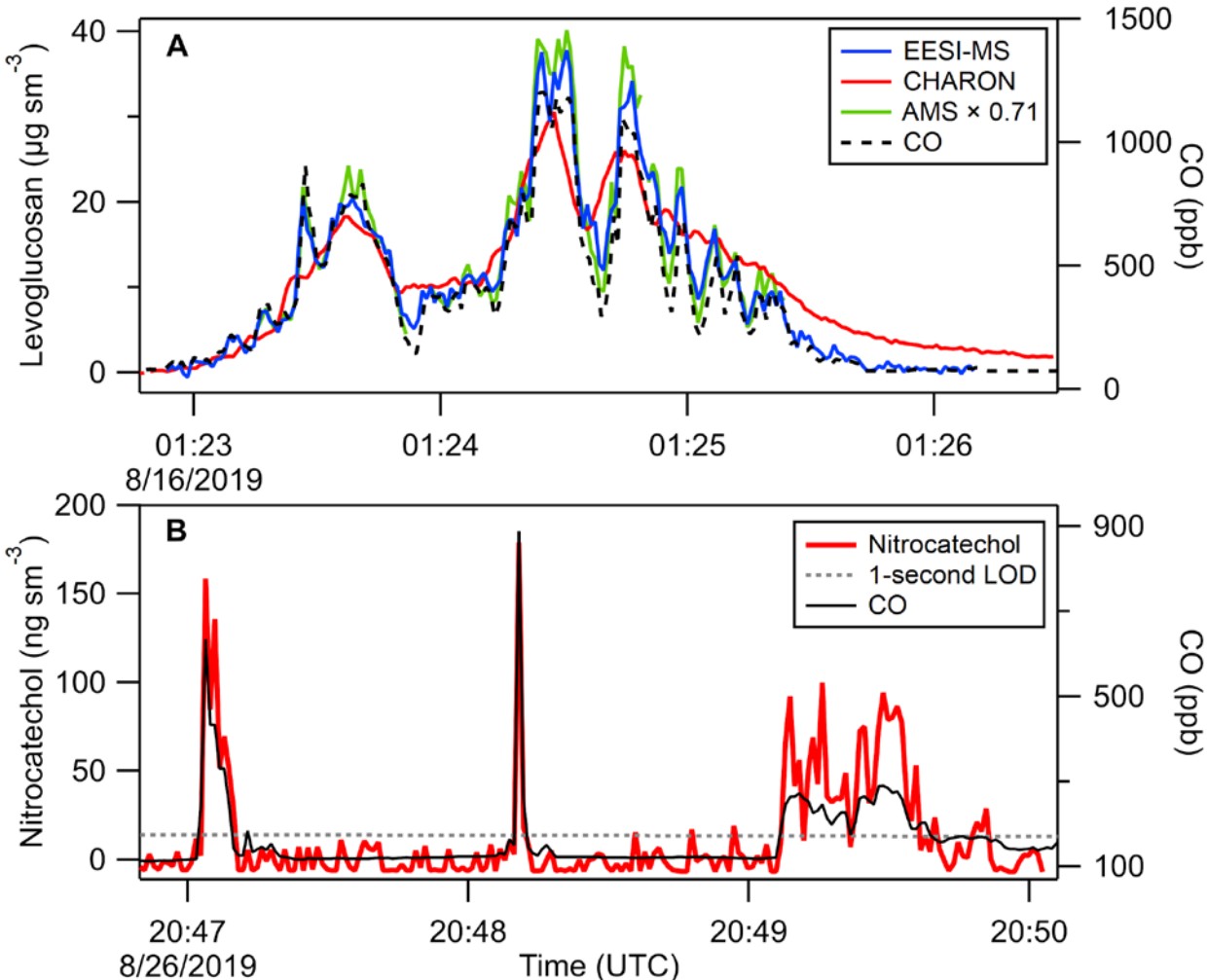

**Figure 5.** Example 1-Hz (A) EESI(+) levoglucosan and (B) EESI(−) nitrocatechol time series from measurements of wildfire smoke aerosol,
including comparison to CHARON PTR-MS and AMS (scaled by a factor of 0.71 to show temporal agreement). Carbon monoxide
measurements are included to show the boundaries and structure of the smoke plumes.

   Calibrated 1-second time series of levoglucosan and nitrocatechol are shown in Fig. 5, demonstrating the fast time response of
airborne EESI-MS. Carbon monoxide measurements are included to illustrate the spatio-temporal boundaries and internal
variability of each smoke plume. In addition to levoglucosan and nitrocatechol, we also quantified the total aerosol EESI-MS
signal, which correlated with AMS OA, as shown in Fig. 6 for both EESI(+) and EESI(−).

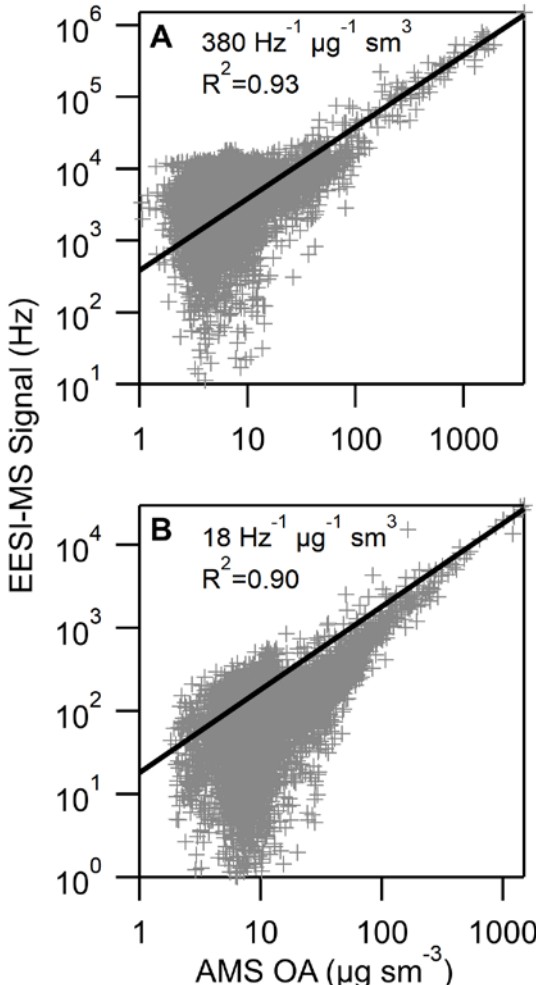

**Figure 6.** Bulk sensitivity of (A) EESI(+) and (B) EESI(−) sampling modes for 1-second data relative to AMS total organic aerosol. Both example flights utilize PCI pressure of 667 mbar.

The coefficient of determination $R^2 \geq 0.9$ for both ion polarities, and the correlation is strongest when OA concentration is above 10 µg sm$^{-3}$. The regression slope of EESI-MS signal vs AMS OA is the bulk OA sensitivity of the EESI-MS. Previous EESI-MS field measurements have carried out levoglucosan calibrations, and so to compare the bulk OA sensitivity, $S_{OA}$, of our airborne EESI-MS to previous EESI-MS field measurements, we normalize each reported OA sensitivity, $S_{OA}$, to that of levoglucosan ($S_{norm} = S_{OA} / S_{Levo}$) (Stefenelli et al. 2019; Qi et al. 2019; Brown et al. 2020). The levoglucosan-normalized

sensitivity of airborne EESI-MS is roughly 60% higher than that of measurements made in Zurich during winter, indicating that biomass burning OA is extracted and ionized with a higher efficiency than urban OA (Fig. S17) (Qi et al. 2019). This is consistent with the high levoglucosan content of BBOA, and is likely impacted by the selection of 3:1 methanol:water mixture as the electrospray working solution for this study. Enhanced sensitivity to BBOA has also been observed using other online

soft-ionization methods (Vogel et al. 2013). Roughly half of EESI-MS signal comes from ten peaks in each polarity, as shown in Fig. S18. The variability in EESI-MS sensitivity to individual compounds varies by over an order of magnitude (Brown et al. 2020; Lopez-Hilfiker et al. 2019), and so it is not clear whether these peaks comprise the majority of OA mass. Identification and calibration of those compounds is planned for future work. Ongoing analysis indicates that the FIREX-AQ EESI-MS dataset contains substantial information on the presence of additional nitroaromatics and organic acids.

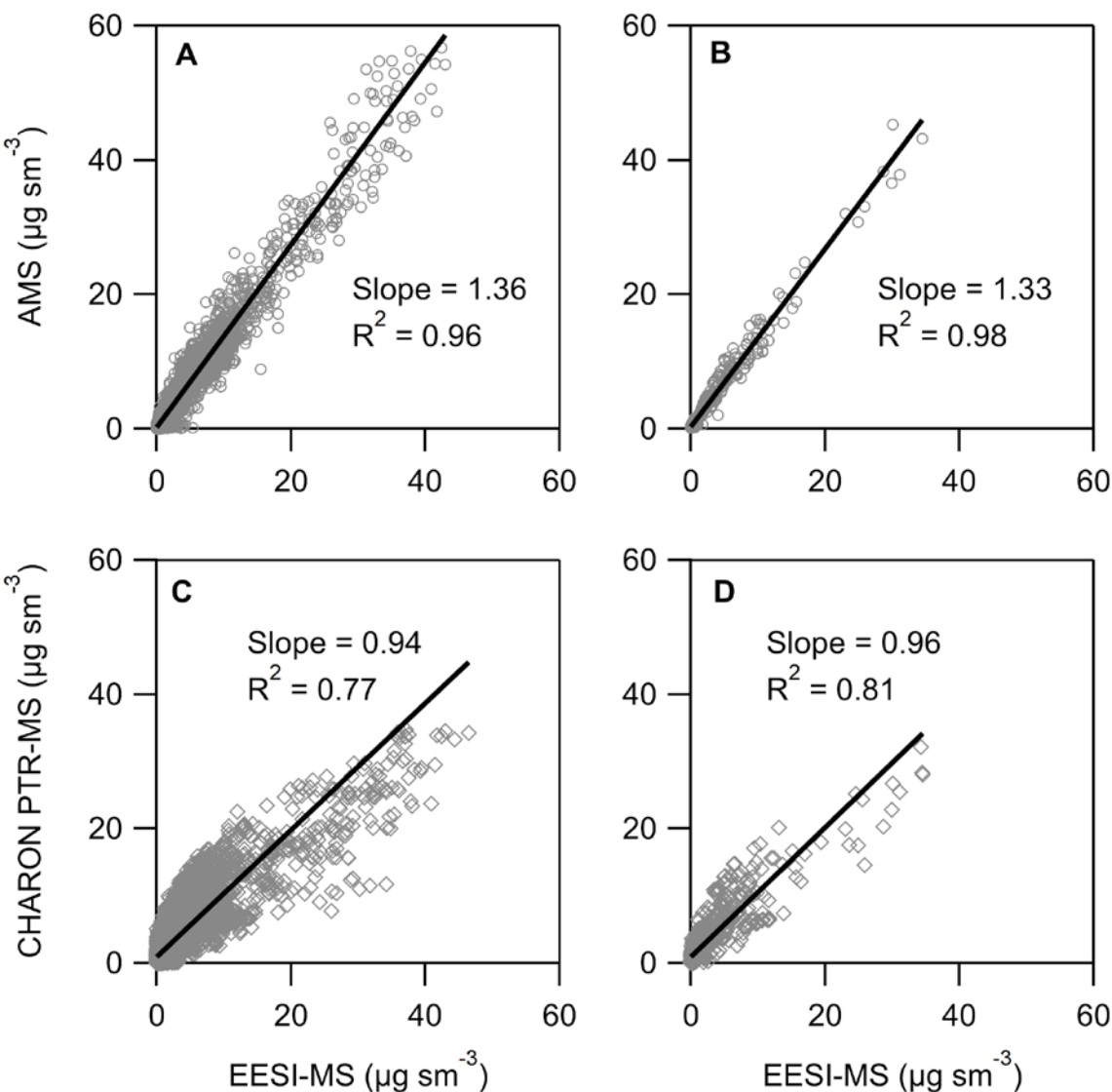

**Figure 7.** Comparison of EESI-MS quantification of levoglucosan ($C_6H_{10}O_5Na^+$) to AMS equivalent levoglucosan at (A) 1 s and (B) 10 s time resolution and to CHARON PTR-MS levoglucosan at (C) 1 s and (D) 10 s time resolution during a single FIREX-AQ flight.

## 3.2 EESI-MS, AMS, and CHARON PTR-MS intercomparison

During one FIREX-AQ research flight, EESI-MS and AMS were flown alongside a CHARON PTR-MS, allowing for an airborne intercomparison of the three instruments. CHARON PTR-MS operates by removing gas-phase organic compounds using a charcoal denuder, concentrating aerosol using an aerodynamic lens, evaporating components of OA using a heated vaporizer at 8 mbar, and detecting those OA components by PTR-MS. More detailed descriptions of the CHARON PTR-MS technique and its airborne operation have been published elsewhere (Piel et al. 2019; Eichler et al. 2015). CHARON PTR-MS and AMS ground measurement intercomparisons have been carried out previously (Müller et al. 2017). Intercomparison of airborne CHARON PTR-MS and airborne EESI-MS with any other aerosol measurement have not been reported before.

EESI(+) was flown during the intercomparison flight, and so levoglucosan concentrations from each instrument at 1 s and 10 s time resolution are compared in Fig. 7. During the intercomparison flight, the CHARON PTR-MS sampled from the University of Hawaii/Langley Aerosol Research Group (UH/LARGE) inlet, which has been shown to have unit transmission efficiency through particle diameters of 1 μm and a 50% cutoff at 4-5 μm (McNaughton et al. 2007; Chen et al. 2011). EESI-MS sampled from the UH/LARGE inlet for part of this flight, and no difference in levoglucosan:CO normalized excess mixing ratios was observed, indicating no difference in the aerosol population sampled by EESI-MS through the HIMIL and UH/LARGE inlets (Fig. S19). Extensive intercomparison of aerosol measurements made using the HIMIL and UH/LARGE inlets are presented elsewhere (Guo et al. 2020). AMS levoglucosan-equivalent concentration is calculated from the fractional intensity of the ion $C_2H_4O_2^+$ ($f_{C2H4O2}$) in ambient OA spectra, the total OA concentration, and $f_{C2H4O2}$ of levoglucosan standards analyzed throughout the campaign during EESI-MS calibrations (Aiken et al. 2009). A subtraction of the contribution of background OA to AMS $C_2H_4O_2^+$ signal is performed before calculating the AMS estimated levoglucosan concentration (Cubison et al., 2011). Because the BBOA concentrations were much larger than the background OA, this subtraction is very minor.

The AMS $C_2H_4O_2^+$ ion has been shown previously to be a marker for anhydrosugars in biomass burning OA (Alfarra et al. 2007; Cubison et al. 2011; Aiken et al. 2009). Contribution from other compounds (including organic acids and sugars) to AMS $C_2H_4O_2^+$ signal has been shown to lead to a higher concentration for AMS levoglucosan-equivalent than for levoglucosan (Aiken et al. 2009; Lee et al. 2010; Zhao et al. 2014; Fortenberry et al. 2018). This trend is observed in Fig. 7 and Fig. S20, where EESI-MS and CHARON PTR-MS levoglucosan concentrations are lower than AMS levoglucosan concentrations by 26% and 34% (calculated relative to AMS levoglucosan). The published ratios of AMS levoglucosan-equivalent to direct measurements of levoglucosan are variable, and the slope of 1.36 observed here is within the previously-reported range, shown in Fig. S21. While this is within the combined uncertainty of these instruments, is also consistent with a ground intercomparison of AMS and CHARON PTR-MS where CHARON PTR-MS levoglucosan was 30% lower than AMS levoglucosan (Müller et al. 2017). As shown in Fig. 7, regression of levoglucosan concentrations measured by EESI-MS and CHARON PTR-MS give a slope of 0.94, $R^2 = 0.77$, which is within the uncertainty of both instruments (EESI-MS 24%, CHARON PTR-MS 30%). Comparing the 1-Hz time series of each instrument (Fig. 5) shows that AMS and EESI-MS respond faster than CHARON

PTR-MS to changes in plume concentration (as indicated by CO concentration). However, due to the sampling arrangements during FIREX-AQ, the sampling line connecting the CHARON PTR-MS to the UH/LARGE inlet had a residence time of 4 s, increasing the sorptive capacity of the CHARON PTR-MS inlet and potentially contributing to the slower time response observed here (Pagonis et al. 2017; Deming et al. 2019). The impact of this inlet effect on the intercomparison can be reduced

(regression slope = 0.96, $R^2$ = 0.81) by increasing the time averaging from 1 s to 10 s, as shown in Fig. 7. Levoglucosan excess mixing ratios with excess CO for EESI-MS, CHARON PTR-MS, and AMS are presented in Fig. 8, showing the same trends as the concentration data discussed above. Excess mixing ratios are determined by subtracting the background concentration of each compound from the in-plume average. Background concentrations were determined by computing 60-second averages before and after each plume transect and interpolating across the plume transect.

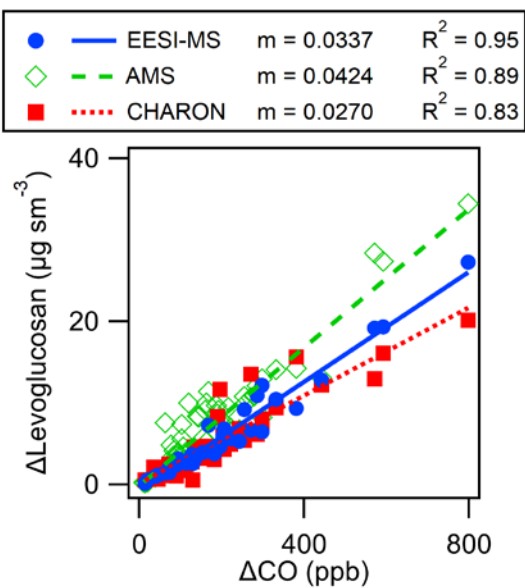

**Figure 8.** Comparison of 1-minute EESI-MS and CHARON PTR-MS excess levoglucosan, and AMS excess levoglucosan-equivalent vs excess CO for a single FIREX-AQ flight. Excess levoglucosan or CO is determined by subtracting the background concentration from the in-plume average concentration.

## 4 Conclusions

We deployed an EESI-MS onboard the NASA DC-8 aircraft during FIREX-AQ and quantified levoglucosan and nitrocatechol concentrations in biomass burning organic aerosol with 1 s time resolution. These measurements required optimization of EESI-MS working solution to allow for operation at pressures as low as 360 mbar, precise control of electrospray capillary position, and flight-day calibrations. Characterization of EESI-MS sensitivity using monodisperse aerosol showed no size dependence for particles smaller than 400 nm in diameter, and no matrix effects were detected for added organic compounds

or inorganic salts. Comparison with previously published EESI-MS bulk OA sensitivities adds support to the idea put forth in those studies that EESI-MS bulk sensitivity varies with OA chemical composition, although far less than for individual species.

EESI-MS levoglucosan concentrations were consistent with those measured using AMS and CHARON PTR-MS, differing by 6% (CHARON PTR-MS) and 30% (AMS). Taken together these results demonstrate the ability to use EESI-MS for fast and accurate quantification of organic aerosol composition onboard aircraft platforms.

## Data Availability

FIREX-AQ data for EESI-MS and all supporting measurements are publicly available in the NASA Data Archive, doi: 10.5067/SUBORBITAL/FIREXAQ2019/DATA001

## Acknowledgements

This work was supported by NASA grants 80NSSC18K0630 and 80NSSC19K0124, as well as a Cooperative Institute for Research in Environmental Sciences (CIRES) Innovative Research Program (IRP) grant. We thank Felipe Lopez-Hilfiker and the EESI-MS users community for useful discussions and support during the field phase of this research. We thank the CIRES Integrated Instrument Development Facility for their work making the EESI-MS flight-ready. We thank Anne Handschy, the crew of the DC-8, the Ames Earth Science Project Office, and FIREX-AQ leadership for support during FIREX-AQ. We thank Glenn Diskin, Joshua DiGangi, John Nowak, and the DACOM instrument team for the CO measurements used here. The CHARON PTR-MS instrument was partly funded by the Austrian Federal Ministry for Transport, Innovation and Technology (bmvit) through the Austrian Space Applications Programme (ASAP) of the Austrian Research Promotion Agency (FFG). CHARON PTR-MS instrumental support came from Ionicon Analytik; Tomas Mikoviny and Markus Müller provided technical assistance. Felix Piel received funding from the European Union's Horizon 2020 research and innovation program under grant agreement no. 674911 (IMPACT EU ITN).

## Author Contributions

DP, PCJ, DAD, and JLJ designed the experiment and wrote the paper; HG, MKS, WLB, BAN, KS, AL, FP, LT, AW, MMC, GIG, HSH, RHM, DST, CW, EBW collected and analyzed data; HS and DST developed software; and JEK contributed to EESI optimization. All authors reviewed and provided comments for the paper.

## Competing Interests

Jordan Krechmer, Harald Stark, and Benjamin Nault work for Aerodyne Research Inc., which has commercialized the EESI-TOF-MS instrument for geoscience research. Felix Piel works for Ionicon Analytik, which has commercialized the CHARON PTR-MS instrument. David Thomson is the founding partner of Original Code Consulting, which has commercialized the

MICAS-X software. Armin Wisthaler profits from a license agreement (CHARON inlet) between the University of Innsbruck and Ionicon Analytik.

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
