# Peer review of "Airborne Extractive Electrospray Mass Spectrometry Measurements of the Chemical Composition of Organic Aerosol"

_Atmospheric Measurement Techniques, 2020_

## Referee Comment (RC1) · Alexander Vogel (Referee) · 12 Nov 2020

Pagonis et al. show airborne measurements with the extractive electrospray ionization mass spectrometer (EESI-MS) during the FIREX-AQ campaign. They describe quantitative measurements of the biomass burning markers levoglucosan and nitrocatechol in the condensed (aerosol) phase using the positive and negative EESI mode. A careful characterization of inlet losses and particle-size-dependencies of the EESI extraction is presented. A quantitative comparison to an AMS and a CHARON-PTR-MS dataset of levoglucosan (at 1 Hz acquisition) during BB plume intersects impressively demonstrates the agreement between the different techniques. An overestimation of

the levoglucosan AMS signal appears plausible, since oligomeric sugars in BB aerosol can fragment and contribute to m/z 60 in AMS spectra. Oligomeric sugars might appear as intact molecules in the soft-ionization instruments, and thus explain the bias.

The quality of the graphs is very good and the language is fluent and precise. Overall, I can recommend the paper to be published in AMT after addressing the following minor comments:

Minor comments:

Have you considered to check the negative spectra for other nitroaromatics in the BB plumes, e.g. nitrophenol and di-nitro-aromatics? To extract new information from the chemical composition of BB plumes through soft-ionization MS (in addition to what the AMS is already able to show), I think it is important to identify other organic markers that tell something about the origin, age, volatility or multiphase processing of fire plumes. It might be out of the scope of this technical paper, but some motivational words about why we need soft-ionization MS should appear in the introduction.

l. 110: Switching a valve –> low pressure transient –> loss of electrospray? Is this an issue?

l. 120 and throughout the manuscript: please use the minus sign instead of a hyphen in EESI($-$) and negative ions (e.g. (C2H3O2$-$) in l.121.

l. 154: What is the maximum delta T between ambient and aircraft cabin during the campaign? Aren't higher losses expected for aerosols at higher altitudes (colder conditions), since intermediate-volatile compounds might expel levoglucosan into the gas phase when they evaporate during sampling? Can this be an issue?

l. 173: 100 ppm of NaI appears to me as a high concentration of a non-volatile salt, which mass spectrometrists usually like to avoid blowing into the MS. This working solution has been used in past EESI studies, but also caused to my knowledge trouble during field experiments. Can you please report how robust is the electrospray against

salt deposition on the tip of the EESI needle? Would a volatile ammonium salt (e.g. ammonium acetate) be an alternative to NaI?

l. 174: Doping the working solution with formic acid in negative ionization mode is questionable. Formic acid is a stronger acid than nitrocatechol (the analyte in the negative ionization mode). At low pH, there might be a suppression of the nitrocatechol ion formation due to a high proton concentration. In our lab we tested the sensitivity of Ibuprofen (organic acid) ionized with ESI$(-)$, and we found two orders of magnitude higher sensitivity when leaving formic acid (0.1 % v/v) out of the mobile phase solvents.

However, ESI and EESI are different ionization processes, and it might be correct that under EESI conditions the formation of FA-anions is occurring before collision with the sampled aerosol. Then the low pH of the working solution might not appear problematic. But, if the electrospray droplets hit the sampled aerosol before Coulomb explosion, a low pH of the EESI working solution might suppress the ionization. Please explain or comment.

l. 270: How well is the HR-fit during background measurements of levoglucosan? Can it be that the high LOD for levoglucosan can partly be explained by erroneous peak attribution to levoglucosan from the left shoulder of C8H18O3Na+? Please provide a figure in the SI of the HR fit during a background measurement.

Figure S10 shows for EESI(+) that only a few compounds (<10) have signal-to-background ratios above one, indicating that background correction potentially can introduce a large bias on the signal intensity. I think the authors were very careful in determining the EESI background. However, no figure reports the variability of the background signal between subsequent HEPA-background measurements. I think that such a figure or a table (with the mean and SD of the grey area in Fig. S1 for a set of background measurements) would be beneficial in order to provide the reader an impression of the background variability.

l. 339: Having a larger overall OA sensitivity during BB-episodes has also been

demonstrated by other online soft-ionization methods than EESI: In Vogel, AMT, 2013 (https://doi.org/10.5194/amt-6-431-2013) we showed in figure 5 that during a biomass burning episode we observed an above-average of APCI OA signal compared to AMS OA.

Technical notes

l. 121: The acetate signal in Fig S2 exceeds 100 cps.

l. 144: SESI –> EESI

Figure 4: Numbers on the y-axis of panel B and D are missing.

Figure 4: I assume that the green spectra are the ones that are background-corrected? This should be clear from the legend.

[Figure]

---

## Referee Comment (RC2) · Anonymous Referee #2 · 5 Dec 2020

This work describes results from an aircraft deployment of the EESI-MS instrument over Western U.S. fires. The study obtained high altitude, fast time resolution, soft ionization measurements of particle-phase biomass burning marker compounds. The manuscript is very well written and clear to follow. Details of instrument operation, data processing and interpretation, and comparison with two other measurement methods are included. The manuscript focuses more on the development of a new technique for a new application, and less on the science question of biomass plume composition, aging, and transport. It is my opinion that it is an appropriate body of work for inclusion in AMT.

Here I include several specific comments and questions.

1. Line 53: What happens to aerosol components that don't go into solution with electrospray drops?

2. Also, what happens to any extremely low volatility components that don't evaporate and may remain clustered in capillary transfer? (over the m/z 700 that was recorded here)

3. Do all particle sizes interact/contact with the electrospray drops at the same efficiency/extent? I guess the later Fig3b indicates there is size dependence. What about mixing state dependence, although not tested in this study, would you expect to have core-shell type coated aerosol?

4. Is there any chance of ESI liquid composition concentration drifting through the course of a measurement period? This would of course have potential to impact the starting size of sprayed droplets and perhaps the solubility of analytes.

5. Line 105: Applause for attempting and achieving this challenging operation condition. I'm guessing there is much more data not included here that has been filtered out due to instrument operation state transitions

6. Line 110: Did background signals drift significantly over time?

7. Figure 2: Did you intentionally collect any gas-phase signal in this study, beyond what is shown here? (that is, filtering particles and bypassing denuder)

8. Line 245: Agreed, even if all conditions seem unchanged, the technique can be unpredictable and challenging to establish an identical taylor cone at the spray tip.

9. Line 271: why do you think the background levoglucosan signal was so high? Is it slowly evaporating off of non-heated surfaces in the inlet, that had deposited in past samples and standards? Perhaps a deuterated levoglucosan standard could be used in future?

10. Figure 4: a little confusing which mass spectra are being plotted in black and green, raw – background – background subtracted. Please add description.

11. At this point in the paper I'm forgetting where you even did this study. Maybe adding a map at the beginning (even if in supplement) would be helpful so the reader has that visual memory.

12. Any other ions stand out beyond these two for levoglucosan and nitrocatechol? Any oxy-PAH's that may have been soluble?

13. Figure 6: relationship below 10ug/sm3 of OA seems to deviate.

14. Line 368: AMS $C_2H_4O_2+$ has been observed to also come from organic acids in laboratory aged biomass burning samples, potentially offering an explanation for the higher AMS biomass signal here (Fortenberry et al, 2018, ACP)

---

## Referee Comment (RC3) · Anonymous Referee #3 · 8 Dec 2020

General Comments: The authors present a detailed characterization of the deployment of an EESI-ToF-MS for on-line measurements of biomass burning aerosol particles on the NASA DC-8. The sensitivity, size dependence, and an inter-comparison with both the AMS as well as a CHARON PTR-MS are presented. Overall, the authors are able to quantify and measure the time series for two major biomass burning components: levoglucosan and nitrocatechol. This paper is very well written and clear and it provides detailed discussions of the limitations of all the measurements. I especially appreciate the comparison with off-line HPLC-ESI-HRMS analysis to confirm the assignment of the molecular formulas measured in these flights. Overall, I recommend acceptance after the following minor comments are addressed.

[Figure]

Minor comments: 1. Page 6 second paragraph: "Semivolatile gases are removed by the denuder during sampling to prevent their detection by SESI, which disturbs gas-particle equilibrium, leading to aerosol evaporation inside the inlet." What are the time scales for sampling in the EESI inlet? Would a significant amount of re-equilibration be expected?

2. For negative mode EESI, formic acid was added to the droplets. However, the addition of acids is more common in positive ion mode ESI as it provides additional protons for the analytes. Formic acid can increase the signal in negative ion mode for some systems, but I suspect that is not universal. Were other dopants tested? This may be an area for further characterization on EESI-MS to improve negative ion mode signal for different systems.

3. Figure 1: this can be added to the supplemental, but it would be good to include information on the sizes and distances shown in the figure. Specifically the distance between the electrospray tip and the entrance to the capillary (or a reference for these values if provided elsewhere). As mentioned in the manuscript, focusing the aerosol particles into a smaller volume may improve signal, I also suspect that changing the distance (time) for dissolution and drying/Coulomb explosion to occur will be another variable that would be helpful to optimize in the future.

4. On pages 10-11, the detection limits for levoglucosan are reported with the note that there was variation with the sampling history of the instrument which persisted for hours. Were these same sustained signals observed for levoglucosan calibration runs, or is this signal coming from other components in the biomass burning plumes?

5. For figure 4, I would recommend a small change to the labels as the black trace is labeled "Background" but the caption lists it as "Raw".
* * *

---

## Author Response (AR1)

Response to referee comments for:

**Airborne Extractive Electrospray Mass Spectrometry Measurements of the Chemical Composition of Organic Aerosol**

Pagonis et al. *AMTD*, 2020

We thank the referees of this manuscript for their helpful comments. We appreciate the time and effort that goes into thoughtful reviews. Our responses and revisions to the manuscript are below. Text from referees is in black, our responses are in blue, and **revisions to the manuscript are bolded**.

**Referee 1**

Pagonis et al. show airborne measurements with the extractive electrospray ionization mass spectrometer (EESI-MS) during the FIREX-AQ campaign. They describe quantitative measurements of the biomass burning markers levoglucosan and nitrocatechol in the condensed (aerosol) phase using the positive and negative EESI mode. A careful characterization of inlet losses and particle-size-dependencies of the EESI extraction is presented. A quantitative comparison to an AMS and a CHARON-PTR-MS dataset of levoglucosan (at 1 Hz acquisition) during BB plume intersects impressively demonstrates the agreement between the different techniques. An overestimation of the levoglucosan AMS signal appears plausible, since oligomeric sugars in BB aerosol can fragment and contribute to m/z 60 in AMS spectra. Oligomeric sugars might appear as intact molecules in the soft-ionization instruments, and thus explain the bias. The quality of the graphs is very good and the language is fluent and precise. Overall, I can recommend the paper to be published in AMT after addressing the following minor comments:

Minor comments:
R1.1: Have you considered to check the negative spectra for other nitroaromatics in the BB plumes, e.g. nitrophenol and di-nitro-aromatics? To extract new information from the chemical composition of BB plumes through soft-ionization MS (in addition to what the AMS is already able to show), I think it is important to identify other organic markers that tell something about the origin, age, volatility or multiphase processing of fire plumes. It might be out of the scope of this technical paper, but some motivational words about why we need soft-ionization MS should appear in the introduction.

We agree, and have added the following text to line 53 of the introduction:

**"The soft ionization of EESI-MS allows for quantitative measurements of individual compounds, providing insight into the chemical pathways involved in the formation and evolution of ambient OA that is more detailed and source-specific than what can be achieved with harsher ionization techniques (Qi et al. 2019; Stefenelli et al. 2019; Tong et al. 2020)."**

Referee 1 is correct that detailed study of the evolution of BBOA is beyond the scope of this work. In response to this comment we have added text describing some preliminary identification of additional BBOA components in line 344:

**"Ongoing analysis indicates that the FIREX-AQ EESI-MS dataset contains substantial information on the presence of additional nitroaromatics and organic acids."**

R1.2: l. 110: Switching a valve –> low pressure transient –> loss of electrospray? Is this an issue?

Yes, pressure transients from valve switching can disrupt EESI-MS electrospray. We have incorporated this into our discussion of pressure transients on line 197. That text now reads:

**"Pressure fluctuations of that magnitude are not unique to aircraft sampling: common sources of inlet pressure variability, such as pressure drops from sampling through particle filters or switching a valve, can approach 25 mbar. These fluctuations must be avoided…"**

R1.3: l. 120 and throughout the manuscript: please use the minus sign instead of a hyphen in EESI(−) and negative ions (e.g. (C2H3O2−) in l.121.

We have made this correction throughout the manuscript

R1.4: l. 154: What is the maximum delta T between ambient and aircraft cabin during the campaign? Aren't higher losses expected for aerosols at higher altitudes (colder conditions), since intermediate-volatile compounds might expel levoglucosan into the gas phase when they evaporate during sampling? Can this be an issue?

We agree with Referee 1 that more detail surrounding our calculations of evaporative loss is needed. In particular, the low losses of levoglucosan must be reconciled with the higher evaporated fractions for OA. We have added discussion that clarifies that compounds being lost from OA are higher volatility than levoglucosan, as well as clarifying that the greatest evaporation occurs when both the temperature gradient and OA concentration are high. This text now reads:

**"The residence time from the entrance of the denuder is 0.65 s, and we calculate that levoglucosan and nitrocatechol ($C^*_{298}$ = 13 μg m$^{-3}$ for both; Finewax et al., 2018; May et al., 2012) undergo losses of under 2% inside the inlet for all plume conditions sampled. During FIREX-AQ the total OA evaporation while sampling smoke is estimated as 0-28%, with evaporation being greatest when both OA concentrations and the temperature difference between the DC-8 cabin and ambient air were high (we note that OA evaporation is significantly lower in the AMS inlet, as the inlet residence time is a factor of three shorter than that of the EESI-MS). The plume transect estimated to undergo 28% OA evaporation had an OA concentration of 1760 μg sm$^{-3}$ and a $\Delta T$ of 34 K. The evaporative loss is estimated to be almost entirely due to compounds with $C^*_{298} > 10^4$ μg m$^{-3}$, three orders of magnitude more volatile than levoglucosan and nitrocatechol, which are estimated to undergo evaporation of under 2% in these conditions."**

R1.5: l. 173: 100 ppm of NaI appears to me as a high concentration of a non-volatile salt, which mass spectrometrists usually like to avoid blowing into the MS. This working solution has been used in past EESI studies, but also caused to my knowledge trouble during field experiments. Can you please report how robust is the electrospray against salt deposition on the tip of the EESI needle? Would a volatile ammonium salt (e.g. ammonium acetate) be an alternative to NaI?

We have added the following discussion surrounding EESI dopants to ln 218 addressing this comment, as well as comments R1.6 and R3.2 which cover similar topics:

**"There is significant potential for further investigation and optimization of electrospray dopants for EESI-MS. While use of NaI as an EESI(+) dopant provided sufficiently stable electrospray for 8 h research flights, a more volatile salt such as ammonium iodide may result in less salt deposition on the electrospray capillary and in the electrospray region. This could lead to more stable EESI(+) operation in situations where days of continuous electrospray are needed, such as ambient sampling at a ground site. The use of an acid dopant in negative-polarity electrospray has the potential to suppress the ionization of compounds less acidic than the dopant. As part of this study both formic acid and acetic acid were tested as dopants for EESI(−) and were found to give similar sensitivity for nitrocatechol, despite a difference in acidity between the two dopants. It is possible that higher sensitivity to weakly acidic compounds could be achieved with a more weakly acidic dopant, or no dopant at all."**

R1.6: l. 174: Doping the working solution with formic acid in negative ionization mode is questionable. Formic acid is a stronger acid than nitrocatechol (the analyte in the negative ionization mode). At low pH, there might be a suppression of the nitrocatechol ion formation due to a high proton concentration. In our lab we tested the sensitivity of Ibuprofen (organic acid)

ionized with ESI(−), and we found two orders of magnitude higher sensitivity when leaving formic acid (0.1 % v/v) out of the mobile phase solvents. However, ESI and EESI are different ionization processes, and it might be correct that under EESI conditions the formation of FA-anions is occurring before collision with the sampled aerosol. Then the low pH of the working solution might not appear problematic. But, if the electrospray droplets hit the  sampled aerosol before Coulomb explosion, a low pH of the EESI working solution might suppress the ionization. Please explain or comment.

This is a good point, and we have included our discussion in response to comment 1.5 above.

R1.7: l. 270: How well is the HR-fit during background measurements of levoglucosan? Can it be that the high LOD for levoglucosan can partly be explained by erroneous peak attribution to levoglucosan from the left shoulder of C8H18O3Na+? Please provide a figure in the SI of the HR fit during a background measurement.

We have added the recommended supplementary figure to show that background levoglucosan is resolved from the left shoulder of the $C_8H_{18}O_3Na^+$ peak. We added the following text to line 272:

**"The background levoglucosan signal is resolved from neighboring peaks, as shown in Fig. S9."**

And the following figure to the SI:

[Figure]

**Figure S9. Levoglucosan signal during (A) measurement of instrument background and (B) sampling 50 μg sm$^{-3}$ of smoke aerosol during a single FIREX-AQ research flight. The peak $C_6H_{10}O_5Na^+$ is resolved from the adjacent peaks."**

R1.8: Figure S10 shows for EESI(+) that only a few compounds (<10) have signal-to-background ratios above one, indicating that background correction potentially can introduce a large bias on the signal intensity. I think the authors were very careful in determining the EESI background. However, no figure reports the variability of the background signal between subsequent HEPA-background measurements. I think that such a figure or a table (with the mean and SD of the grey area in Fig. S1 for a set of background measurements) would be beneficial in order to provide the reader an impression of the background variability.

To provide a clear picture into the extent of variability in EESI-MS background we have added a supplementary figure from a representative FIREX-AQ flight, including the mean and standard deviation of background measurements:

[Figure]

**"Figure S10. Raw $C_6H_{10}O_5Na^+$ signal calibrated as levoglucosan for a representative EESI(+) FIREX-AQ flight, along with average values for all background measurements. The increase in background is small following pre-flight calibration and the background prior to calibration is high, indicating that pre-flight calibrations are a minor contributor to background. The decrease in background concentration during high-altitude transits suggests that there may be accumulated levoglucosan in the EESI-MS inlet from the previous flight that slowly evaporates. This is supported by the relationship between the intensity of the background signal and instrument temperature. The small contribution of pre-flight calibrations to instrument background could be lessened through use of an isotopically labeled calibration standard. The mean and standard deviation of the background measurements are $1.2 \pm 0.3$ µg m$^{-3}$."**

We have also added reference to this figure and discussion on line 274:

**"...higher detection limits observed following sustained sampling of biomass burning OA, persisting for hours (Fig. S10)."**

R1.9: l. 339: Having a larger overall OA sensitivity during BB-episodes has also been demonstrated by other online soft-ionization methods than EESI: In Vogel, AMT, 2013 (https://doi.org/10.5194/amt-6-431-2013) we showed in figure 5 that during a biomass burning episode we observed an above-average of APCI OA signal compared to AMS OA.

We have made reference to this supporting study by adding the following text to line 341:

**"Enhanced sensitivity to BBOA has also been observed using other online soft-ionization methods (Vogel et al. 2013)."**

Technical notes
R1.10: l. 121: The acetate signal in Fig S2 exceeds 100 cps.

We have clarified that it is the increase in acetate signal that does not exceed 100 cps above the background. The line now reads:

**"...EESI-MS acetate signal ($C_2H_3O_2^-$) increases by less than 100 counts s$^{-1}$ during the intercept of a plume..."**

R1.11: l. 144: SESI –> EESI

We are specifically referring to secondary electrospray ionization of gas-phase compounds. To make this more apparent to readers, line 144 now reads:

**"Organic gases are removed by the denuder during sampling to prevent gas-phase ionization by SESI. The removal of semivolatile gases disturbs gas-particle equilibrium, potentially leading to aerosol evaporation inside the inlet."**

R1.12: Figure 4: Numbers on the y-axis of panel B and D are missing.

We have added values to the y-axis of Fig. 4B and 4D:

[Figure]

"**Figure 4. Raw and background-subtracted EESI(+) (A) and EESI(−) (C) spectra while sampling 50 μg sm⁻³ of wildland fire smoke aerosol, and high-resolution mass spectra and peak fits of ions attributed to (B) levoglucosan and (D) nitrocatechol. The peaks shown in (B) and (D) are from the same spectra as panels (A) and (C)."**

R1.13: Figure 4: I assume that the green spectra are the ones that are background-corrected? This should be clear from the legend.

This is correct. We have updated the legend of Fig. 4 to make this clearer, as shown in response to comment R1.12 above.

**Referee 2**

This work describes results from an aircraft deployment of the EESI-MS instrument over Western U.S. fires. The study obtained high altitude, fast time resolution, soft ionization measurements of particle-phase biomass burning marker compounds. The manuscript is very well written and clear to follow. Details of instrument operation, data processing and interpretation, and comparison with two other measurement methods are included. The manuscript focuses more on the development of a new technique for a new application, and less on the science question of biomass plume composition, aging, and transport. It is my opinion that it is an appropriate body of work for inclusion in AMT.

Here I include several specific comments and questions.

R2.1. Line 53: What happens to aerosol components that don't go into solution with electrospray drops?

We have added the following text to line 166 to describe the fate of aerosol components that were not ionized by EESI:

**"Aerosol components that were not ionized by EESI are not focused by the ion optics of the TOF-MS and are pumped away or deposited on an internal surface of the ionization volume or mass spectrometer."**

R2.2. Also, what happens to any extremely low volatility components that don't evaporate and may remain clustered in capillary transfer? (over the m/z 700 that was recorded here)

We have added the following text to line 166 to describe the fate of ionized high-mass components of OA and provided a reference that details an instance where this has occurred:

**"Any ions above this *m/z* are recorded by the detector as part of a subsequent mass spectrum, in an effect known as "TOF wraparound" (Brown et al. 2020)."**

R2.3. Do all particle sizes interact/contact with the electrospray drops at the same efficiency/extent? I guess the later Fig 3b indicates there is size dependence. What about mixing state dependence, although not tested in this study, would you expect to have core-shell type coated aerosol?

Past work on the impact of phase state on EESI-MS quantification is not conclusive, as discussed on lines 258-266 of the AMTD version of the paper. We have added text so that this section more directly addresses Referee 2's question. The section now reads:

**"Here we only tested mixtures that could be generated from a single nebulized aqueous solution, but previous studies have examined the effect of coatings on EESI-MS sensitivity and reported differing results (Kumbhani et al., 2018; Lopez-Hilfiker et al., 2019). It is discussed in Kumbhani et al. (2018) that the large particle size (up to 600 nm) may have been a key factor in the incomplete solvation of their multiphase aerosol particles, which would be consistent with the reduction of EESI-MS sensitivity observed in this study for particles with diameters larger than 400 nm. Additional studies are needed to separate the contributions of particle diameter and particle phase separation to EESI solvation efficiency. The instrument intercomparisons during measurement of wildfire smoke aerosol presented below provide evidence that EESI-MS sensitivity calculated from one-component and two-component calibrant mixtures can be applied to more complex matrices, and that there were no significant phase state limitations on EESI-MS quantification of BBOA during FIREX-AQ."**

R2.4. Is there any chance of ESI liquid composition concentration drifting through the course of a measurement period? This would of course have potential to impact the starting size of sprayed droplets and perhaps the solubility of analytes.

We have added the following text to line 175 to address this question:

**"Electrospray capillaries and the high-voltage electrode were cleaned with methanol prior to entering the working solutions to avoid any contamination, which would have increased instrument background over the course of the campaign. Working solutions were kept sealed to prevent evaporation from affecting the solvent:solute ratio."**

R2.5. Line 105: Applause for attempting and achieving this challenging operation condition. I'm guessing there is much more data not included here that has been filtered out due to instrument operation state transitions

We have added the following to line 294 to describe the data coverage achieved by EESI-MS during FIREX-AQ:

**"EESI-MS data at FIREX-AQ covers 414 out of 538 plume transects (77%). Of those transects with no EESI-MS data, the majority (76 out of 124) are from the three research flights where EESI-MS was flown without a denuder. Excluding those flights, EESI-MS data covers 90% of plume transects. Four percent of FIREX-AQ plume transects occurred above the operational ceiling of the EESI-MS."**

R2.6. Line 110: Did background signals drift significantly over time?

We agree with the Referees that this topic warranted additional discussion. We have provided a figure and discussion on this topic in our response to comment R1.8 above.

R2.7. Figure 2: Did you intentionally collect any gas-phase signal in this study, beyond what is shown here? (that is, filtering particles and bypassing denuder)

We did not. We considered it, but decided to focus our efforts on the particle-phase analysis, given the additional complexities that arise from gas-phase analysis and the significant challenges associated with designing an airborne inlet suitable for simultaneous particle and gas sampling. We have added the following text to line 293 to make it clear to readers that we did not do additional gas-phase sampling beyond what was shown:

**"EESI-MS was flown with a denuder for all other research flights."**

R2.8. Line 245: Agreed, even if all conditions seem unchanged, the technique can be unpredictable and challenging to establish an identical taylor cone at the spray tip.

We thank Referee 2 for this framing of the challenges of quantitative electrospray signals, and have adopted it in the following text, added to line 245:

**"Recalibration is necessary even if all conditions seem unchanged, as the same primary ESI ion signal can arise from electrosprays with different properties."**

R2.9. Line 271: why do you think the background levoglucosan signal was so high? Is it slowly evaporating off of non-heated surfaces in the inlet, that had deposited in past samples and standards? Perhaps a deuterated levoglucosan standard could be used in future?

We agree with Referee 2 that evaporation of levoglucosan from surfaces plays a role. We have included discussion on this topic as part of our response to comment R1.8 above.

R2.10. Figure 4: a little confusing which mass spectra are being plotted in black and green, raw – background – background subtracted. Please add description.

We have updated the labels and description of Figure 4. The updated figure and caption are shown in response to comment R1.12 above.

R2.11. At this point in the paper I'm forgetting where you even did this study. Maybe adding a map at the beginning (even if in supplement) would be helpful so the reader has that visual memory.

We have added a map as Fig. S12. The following text has been modified on line 286:

**"...as part of the FIREX-AQ study (campaign map shown in Fig. S12)."**

And the following figure has been added to the SI:

[Figure]

**"Figure S12. Flight map for the NASA DC-8 during FIREX-AQ, with smoke plume transects shown as markers."**

R2.12. Any other ions stand out beyond these two for levoglucosan and nitrocatechol? Any oxy-PAH's that may have been soluble?

We have modified the text at line 342 to emphasize that other components of OA have not yet been definitively identified. The updated text is presented in the response to comment R1.1 above.

R2.13. Figure 6: relationship below 10 ug/sm3 of OA seems to deviate.

We have made note of this by adding the following text to line 330:

**"The coefficient of determination $R^2 \geq 0.9$ for both ion polarities, and the correlation is strongest when OA concentration is above 10 μg sm$^{-3}$."**

R2.14. Line 368: AMS C2H4O2+ has been observed to also come from organic acids in laboratory aged biomass burning samples, potentially offering an explanation for the higher AMS biomass signal here (Fortenberry et al, 2018, ACP)

We have incorporated this supporting study into our discussion of AMS $C_2H_4O_2^+$. The sentence at line 367 now reads:

**"Contribution from other compounds in BBOA (including organic acids and sugars) to AMS $C_2H_4O_2^+$ signal has been shown to lead to a higher concentration for AMS levoglucosan-equivalent than for levoglucosan (Aiken et al. 2009; Lee et al. 2010; Zhao et al. 2014; Fortenberry et al. 2018)."**

We have also added additional context around the variability in the ratio of AMS levoglucosan-equivalent to direct measurements of levoglucosan to line 371:

**"The published ratios of AMS levoglucosan-equivalent to direct measurements of levoglucosan are variable, and the slope of 1.36 observed here is within the previously-reported range, shown in Fig. S21 (Aiken et al. 2009; Lee et al. 2010)."**

We have also added the following figure to the SI:

[Figure]

**"Figure S21. Comparison of AMS levoglucosan-equivalent to levoglucosan measured by EESI-MS (this work), CHARON PTR-MS (this work), high-performance anion-exchange chromatography (Lee et al. 2010), and GC-MS (Aiken et al. 2009)."**

To clarify the contribution of non-BBOA compounds to the AMS $C_2H_4O_2^+$ signal we also added the following text to line 366:

**"A subtraction of the contribution of background OA to AMS $C_2H_4O_2^+$ signal is performed before calculating the AMS estimated levoglucosan concentration (Cubison et al., 2011). Because the BBOA concentrations were much larger than the background OA, this subtraction is very minor."**

**Referee 3**

General Comments: The authors present a detailed characterization of the deployment of an EESI-ToF-MS for on-line measurements of biomass burning aerosol particles on the NASA DC-8. The sensitivity, size dependence, and an inter-comparison with both the AMS as well as a CHARON PTR-MS are presented. Overall, the authors are able to quantify and measure the time series for two major biomass burning components: levoglucosan and nitrocatechol. This paper is very well written and clear and it provides detailed discussions of the limitations of all the measurements. I especially appreciate the comparison with off-line HPLC-ESI-HRMS analysis to confirm the assignment of the molecular formulas measured in these flights. Overall, I recommend acceptance after the following minor comments are addressed.

R3.1. Page 6 second paragraph: "Semivolatile gases are removed by the denuder during sampling to prevent their detection by SESI, which disturbs gas-particle equilibrium, leading to aerosol evaporation inside the inlet." What are the time scales for sampling in the EESI inlet? Would a significant amount of re-equilibration be Expected?

We agree with the Referees that additional detail on the extent of OA evaporation is needed. We have expanded the text at line 150, as shown in response to comment R1.4 above.

R3.2. For negative mode EESI, formic acid was added to the droplets. However, the addition of acids is more common in positive ion mode ESI as it provides additional protons for the analytes. Formic acid can increase the signal in negative ion mode for some systems, but I suspect that is not universal. Were other dopants tested? This may be an area for further characterization on EESI-MS to improve negative ion mode signal for different systems.

We have included our description of other EESI(−) dopants tested as part of this study and discussion of the potential for further optimization of EESI in response to comment R1.5.

R3.3. Figure 1: this can be added to the supplemental, but it would be good to include information on the sizes and distances shown in the figure. Specifically the distance between the electrospray tip and the entrance to the capillary (or a reference for these values if provided elsewhere). As mentioned in the manuscript, focusing the aerosol particles into a smaller volume may improve signal, I also suspect that changing the distance (time) for dissolution and drying/Coulomb explosion to occur will be another variable that would be helpful to optimize in the future.

We have added text to line 164 describing this key dimension and the possibility of adjusting the electrospray-heated capillary distance to further optimize airborne EESI-MS:

**"The heated capillary is 4 ± 0.5 mm from the electrospray tip, depending on the optimized electrospray position. It is possible that future work optimizing this distance (and thereby time for droplet evaporation) may assist in achieving stable electrospray at lower pressures than those used in this study."**

R3.4. On pages 10-11, the detection limits for levoglucosan are reported with the note that there was variation with the sampling history of the instrument which persisted for hours. Were these same sustained signals observed for levoglucosan calibration runs, or is this signal coming from other components in the biomass burning plumes?

We agree with the Referees that this topic warrants additional discussion. We have provided a figure and discussion on this topic in our response to comment R1.8 above.

R3.5. For figure 4, I would recommend a small change to the labels as the black trace is labeled "Background" but the caption lists it as "Raw".

We have updated the labels of Figure 4. The figure and caption are shown in response to comment R1.12 above.

[revised manuscript text omitted]

**Chemical Purities and Suppliers**

The following chemicals were used in this study: acetonitrile (Thermo Scientific, UHPLC-MS grade); ammonium nitrate (Fisher Scientific, Certified ACS); ammonium sulfate (Fisher Scientific, Certified ACS); formic acid (Ricca, >99%); levoglucosan (CHEM-IMPEX International, >99%); methanol (Sigma-Aldrich, HPLC > 99.9%); 4-nitrocatechol (Sigma-Aldrich, 97%); pinonic acid (Sigma-Aldrich, 98%); sodium iodide (Acros Organics, 99.999% trace metals basis); and water (Thermo Scientific, UHPLC-MS grade).

[Figure]

35  **Figure S1.** Response time of EESI-MS to background measurements with zero air (A), and response time during fast plume crossings in EESI(+) (B) and EESI(-) (C) operating modes. CO data is shown at 5 Hz to show plume boundaries and structure. EESI-MS response is sufficiently fast for reporting data at 1 Hz.

[Figure]

**Figure S2.** Quantification of EESI-MS denuder efficiency for removing gas-phase VOCs. EESI(-) acetate signal during wildfire smoke sampling with the carbon denuder in the inlet (A) and with no denuder in place (B). Comparison to PTR-MS measurements of acetic acid are included to show that similar concentrations of VOC were sampled in both example time series. Identical to Fig. 2, but with y-scaling in (A) adjusted to show the weak correlation of EESI(-) acetate with PTR-MS acetic acid.

**Denuder Sorptive Capacity**

Denuder sorptive capacity was estimated from the estimated geometric surface area of the denuder (207 cm$^2$), an assumed roughness factor of 200, and an assumed surface site density of $10^{14}$ sites cm$^{-2}$. Roughness factor was estimated from experiments measuring total VOC desorbed during bakeout of a similar denuder using PTR-MS (Bakker-Arkema and Ziemann, Personal Communication). This gives a sorptive capacity of $6 \times 10^{18}$ molecules for the entire denuder, corresponding to a capacity of 3 ppm hr at 1 atm, 298 K, and 1 L min$^{-1}$.

[Figure]

50 **Figure S3.** EESI-MS inlet residence times at PCI pressures of 467 mbar and 667 mbar as a function of sampling altitude, colored to show
the contribution of each inlet subassembly.

[Figure]

**Figure S4.** Calculation of particle transmission through the EESI-MS inlet as a function of sampling altitude and particle geometric diameter.

[Figure]

**Figure S5.** Estimated EESI-MS inlet transmission efficiency at a flight altitude of 1 km with respect to particle impaction in tubing bends, diffusion, gravitational settling, aspiration, and impaction behind the PCI critical orifice.

[Figure]

**Figure S6.** Estimated EESI-MS inlet transmission as a function of particle geometric diameter at representative flight altitudes and during ground calibrations. Dashed lines for EESI-MS cases show transmission for all inlet components besides the PCI critical orifice. AMS aircraft instrument transmission (Guo et al., 2020) and campaign-averaged FIREX-AQ aerosol volume and number distributions measured by laser aerosol spectrometer (LAS) are overlaid.

**Critical orifice loss calculations**

Particle losses from sampling through the PCI critical orifice are parameterized using the data of Chen et al. (2007). Large particle losses are dominated by deposition to the tube after the orifice. We parameterize the transmission efficiency (*TE*) in Chen et al. (2007) by Stokes number (*Stk*) using the sigmoidal function shown in Eqn. S1:

$$TE = 1 - \left(1 + e^{\frac{0.715 \times Stk^{0.5}}{0.16}}\right)^{-1}$$

(S1)

The critical orifice losses are roughly coincident with bend losses in the inlet, giving a 50% cutoff geometric diameter of ≈1 μm at all altitudes. During ground calibrations, the inlet is operated at a lower flow rate, reducing particle impaction losses in tubing bends. The critical orifice is then the dominant loss process for particles larger than 300 nm during ground operation, as shown in Fig. S6.

[Figure]

**Figure S7.** Photograph of the EESI-MS pressure-controlled inlet, including the linear actuator controlling the electrospray capillary position.

75

[Figure]

**Figure S8.** EESI(+) signal for primary electrospray ions and levoglucosan aerosol standard as a function of electrospray capillary position. Position 0 mm corresponds to the smallest distance between the electrospray capillary and sampling capillary where EESI-MS signal is still obtained. The position scan presented started at 3 mm and moved the electrospray capillary towards the sampling capillary (pushing electrospray capillary further into the spray region; downward in Fig. S7).

80

[Figure]

**Figure S9.** Levoglucosan signal during (A) measurement of instrument background and (B) sampling 50 μg sm$^{-3}$ of smoke aerosol during a single FIREX-AQ research flight. The peak C$_6$H$_{10}$O$_5$Na$^\pm$ is resolved from the adjacent peaks.

[Figure]

**Figure S10.** Raw $C_6H_{10}O_5Na^{\pm}$ signal calibrated as levoglucosan for a representative EESI(+) FIREX-AQ flight, along with average values for all background measurements. The increase in background is small following pre-flight calibration and the background prior to calibration is high, indicating that pre-flight calibrations are a minor contributor to background. The decrease in background concentration during high-altitude transits suggests that there may be accumulated levoglucosan in the EESI-MS inlet from the previous flight that slowly evaporates. This is supported by the relationship between the intensity of the background signal and instrument temperature. The small contribution of pre-flight calibrations to instrument background could be lessened through use of an isotopically labelled calibration standard. The mean and standard deviation of the background measurements are $1.2 \pm 0.3$ µg m$^{-3}$.

[Figure]

**Figure S11.** Histograms for EESI(+) levoglucosan and EESI(–)(–) nitrocatechol detection limits at 1 second time resolution at (A, C) 467 mbar (A, C) and (B, D) 667 mbar (B, D) PCI pressure for all EESI-MS ambient sampling during FIREX-AQ. The different modes visible in (B) and (DC) demonstrate the difference in performance achieved with different electrosprays. The detection limit of each electrospray varies slightly with sampling history, but the spray-to-spray variability can be larger, again demonstrating the importance of calibrating each electrospray used.

[Figure]

**Figure S12.** Flight map for the NASA DC-8 during FIREX-AQ, with smoke plume transects shown as markers.

105

[Figure]

**Figure S13.** Signal-to-background ratio for the (A) EESI(+) and (B) EESI(-) mass spectra shown in Fig. 4. Both spectra are of 50 µg sm$^{-3}$ of biomass burning OA.

[Figure]

110

**Figure S14.** Allan standard deviation for (A) EESI(+) (A) and (B) EESI(-) (B)(−) primary spray ions and analyte signals, calculated from extended in-flight measurements of instrument background. Signals with significant contribution from the electrospray background (e.g. primary spray ions and levoglucosan) show minima in Allan deviation near 20 s of averaging, while low-background analyte signals follow ideal counting statistics (N$^{-\frac{1}{2}}$).

115

[Figure]

**Figure S15.** HPLC-ESI(–)(−) QE Orbitrap-MS ultrahigh-resolution $C_6H_4NO_4^-$ chromatogram of a FIREX-AQ filter extract showing a single peak matching the retention time of a 4-nitrocatechol standard. Inset: average mass spectrum of the full chromatogram, showing no ESI-MS interference peaks at $m/z$ 154.

[Figure]

**Figure S16.** HPLC- ESI(+) QE Orbitrap-MS ultrahigh-resolution $C_6H_{10}O_5^+$ chromatogram of a FIREX-AQ filter extract showing a single peak matching the weakly-retained retention time of a levoglucosan standard. Inset: average mass spectrum of the full chromatogram. The background peak $C_8H_{18}O_3Na^+$ is present in both ESI-MS and EESI-MS, and is resolved from $C_6H_{10}O_5^+$ by EESI-MS, as shown in Fig. 4. and Fig. S9.

[Figure]

**Figure S17.** Comparison of EESI(+) sensitivity in this study to past EESI-MS field measurements at HOMEChem (Brown et al. 2020), as well as Zurich in summer and winter (Stefenelli et al. 2019; Qi et al. 2019). Sensitivities shown are normalized to levoglucosan sensitivity, which was used for calibration across all four studies, to account for variable electrospray region pressures and other instrument parameters. The Zurich summer and HOMEChem studies utilized a 1:1 acetonitrile:water electrospray working solutions, while the Zurich winter study utilized 1:1 methanol:water and this study utilized 3:1 methanol:water. The varying amounts of levoglucosan present track with the relative sensitivities, so it is not possible to separate the contribution of the working solution to the bulk OA sensitivity from the data available.

[Figure]

**Figure S18.** Cumulative fraction of OA signal as a function of the number of EESI-MS peaks giving positive signal during the research flights shown in Fig. 6. Roughly ten peaks give half the signal in both MS modes. Since EESI-MS sensitivity can vary by orders of magnitude between compounds (Lopez-Hilfiker et al. 2019; Brown et al. 2020), it is not clear whether these peaks comprise the majority of OA mass.

[Figure]

140

**Figure S19.** (A) Comparison of EESI-MS levoglucosan:CO 1 Hz emission ratios for the Sheridan fire when sampling through the HIMIL inlet and through the UH/LARGE inlet during the August 15th, 2019 research flight, and (B) the laser aerosol spectrometer size (LAS) distributions measured through the UH/LARGE inlet during the two sampling periods described in A. Identical emission ratios are measured through the two inlets. The calculated EESI-MS particle transmission at a pressure altitude of 4.7 km is shown to demonstrate that the volume

145 distributions measured by LAS during the two sampling periods do not show any appreciable particle volume beyond the EESI-MS cutoff diameter.

[Figure]

**Figure S20.** Comparison of 1-second CHARON PTR-MS quantification of levoglucosan to AMS levoglucosan during a single FIREX-AQ
150    flight.

[Figure]

**Figure S21.** Comparison of AMS levoglucosan-equivalent to levoglucosan measured by EESI-MS (this work), CHARON PTR-MS (this work), high-performance anion-exchange chromatography of filter samples (Lee et al. 2010), and GC-MS (Aiken et al. 2009).